# FSTLLM: Spatio-Temporal LLM for Few Shot Time Series Forecasting

**Yue Jiang** [1 2]  **Yile Chen\*** [1]  **Xiucheng Li** [3]  **Qin Chao** [1 2]  **Shuai Liu** [1]  **Gao Cong** [1]

## Abstract

Time series forecasting fundamentally relies on accurately modeling complex interdependencies and shared patterns within time series data. Recent advancements, such as Spatio-Temporal Graph Neural Networks (STGNNs) and Time Series Foundation Models (TSFMs), have demonstrated promising results by effectively capturing intricate spatial and temporal dependencies across diverse real-world datasets. However, these models typically require large volumes of training data and often struggle in data-scarce scenarios. To address this limitation, we propose a framework named Few-shot Spatio-Temporal Large Language Models (FSTLLM), aimed at enhancing model robustness and predictive performance in few-shot settings. FSTLLM leverages the contextual knowledge embedded in Large Language Models (LLMs) to provide reasonable and accurate predictions. In addition, it supports the seamless integration of existing forecasting models to further boost their predicative capabilities. Experimental results on real-world datasets demonstrate the adaptability and consistently superior performance of FSTLLM over major baseline models by a significant margin. Our code is available at: https://github.com/JIANGYUE61610306/FSTLLM.

## 1. Introduction

Multivariate time series forecasting methodologies are essential for assisting human experts in decision-making, resource allocation, and scheduling across various industries. Accurate forecasting requires precise modeling on two dimensions: temporal dependency, which captures the dynamic evolution of data over time, and spatial correlation (or channel correlation), which describes the interdependencies between different time series. These correlations typically emerge from mutual influences between time series, such as those driven by physical proximity (Li et al., 2017) or functional similarity (Bai et al., 2020; Wu et al., 2020; Chen et al., 2024). In practice, spatial correlations in multivariate series are not explicit, and effectively modeling such latent correlations is critical for more accurate predictions (Shao et al., 2022b; Zheng et al., 2020; Han et al., 2021). Traditional approaches, including statistical models like ARIMA (Williams & Hoel, 2003) and VAR (Zivot & Wang, 2006), as well as machine learning methods such as SVR (Drucker et al., 1996) and Gaussian Processes (Xie et al., 2010), focus on modeling short-term temporal dynamics. Recent studies, particularly Transformer-based frameworks (Zhou et al., 2021; Wu et al., 2021; Zhang & Yan, 2023; Woo et al., 2022; Cirstea et al., 2022), have significantly enhanced forecasting performance by effectively modeling the long-term temporal dependencies. Besides, emerging approaches for Time Series Foundation Models (TSFMs), such as GPT4TS (Zhou et al., 2023) and Time-LLM (Jin et al., 2024), explore the adaptation of Large Language Models (LLMs) to time series forecasting through partial fine-tuning strategies. In parallel, Spatio-Temporal Graph Neural Networks (STGNNs) have been developed for scenarios where each channel of a multivariate time series is associated with spatial attributes that exhibit mutual influence (Guo et al., 2019; 2022; Shao et al., 2022a; Yu et al., 2017; Song et al., 2020; Shao et al., 2022b; Shang et al., 2021; Miao et al., 2024b; Wang et al., 2022). These models are designed to jointly model spatial and temporal dependencies, enabling a more holistic modeling for correlated time series data.

Despite these advancements, several limitations persist. First, these models typically require substantial amounts of training data to effectively learn complex spatio-temporal correlations inherent in time series. However, collecting such large-scale data can be time-consuming and resource-intensive in practice, potentially requiring months to accumulate sufficient quantities for training. This constraint poses a challenge in many real-world scenarios, where timely and sufficient data may not be readily available, thereby limiting the effectiveness of these data-hungry models. Second, both STGNNs and TSFMs normalize time

[1]College of Computing and Data Science, Nanyang Technological University, Singapore [2]DAMO Academy, Alibaba group, Singapore [3]School of Computer Science and Technology, Harbin Institute of Technology (Shenzhen), China. Correspondence to: Yile Chen <yile001@e.ntu.edu.sg>.

*Proceedings of the 42$^{nd}$ International Conference on Machine Learning*, Vancouver, Canada. PMLR 267, 2025. Copyright 2025 by the author(s).

series data into numerical vectors and update model parameters without integrating the contextual knowledge of real-world factors, such as geographical information, urban contexts, and human behavioral patterns. Solely relying on numerical inputs limits the models' ability to capture these external factors, which are often crucial for improving forecasting accuracy across diverse domains. In contrast, LLMs demonstrate strong capabilities in common sense reasoning (Zhao et al., 2023), making them particularly effective for integrating domain-specific and contextual knowledge via in-context learning (Min et al., 2022) and fine-tuning. Furthermore, LLMs exhibit robust performance in few-shot and zero-shot learning settings (Brown et al., 2020; Kojima et al., 2022), which are highly advantageous in data-scarce forecasting scenarios. These two characteristics motivate the exploration of LLMs for few-shot time series forecasting. While some recent LLM-based time series forecasting methods (Jin et al., 2024; Zhou et al., 2023; Chang et al., 2023; Pan et al., 2024) have made initial strides toward this direction, they reply on fine-tuning LLMs using purely numerical inputs (Zhou et al., 2023; Chang et al., 2023; Liu et al., 2024a), or concatenate general task-specific instructions as prompts (Jin et al., 2024). However, these strategies fall short of fully leveraging the contextual understanding of real-world factors inherent in different time series data. Additionally, some studies have reported that replacing the LLM component from these methods with a sample layer does not degrade performance (Tan et al., 2024).

To address the limitations of existing approaches in data-scarce settings, we propose Few-shot Spatio-Temporal Learning with Large Language Models (FSTLLM), a novel and flexible framework designed to enhance forecasting performance by integrating diverse STGNN backbones with the contextual reasoning capabilities of LLMs. FSTLLM leverages the rich contextual knowledge embedded in LLMs to improve the modeling of both spatial and temporal dependencies, thus substantially reducing reliance on large volumes of training data. Specifically, it treats each time series channel as a node and employs an LLM-Enhanced Graph Construction Module that generates node-specific contextual embeddings from textual contents. These embeddings are then used to construct an adjacency matrix that captures semantically meaningful spatial correlations, which are subsequently used as input to an STGNN backbone for spatio-temporal modeling. To further refine the predictions produced by the STGNN, FSTLLM incorporates a Domain Knowledge Injection Module that fine-tunes the LLM using carefully designed prompts. These prompts integrate domain-specific knowledge, node-level spatio-temporal context, and predictive cues derived from the STGNN backbone, enabling the model to reason over both structured and contextual information. Compared to prior LLM-based forecasting methods, FSTLLM introduces a more expressive

and context-aware prompt design, allowing the model to incorporate reasoning about spatial and temporal contexts and the initial predictions from STGNN backbones for few-shot settings. Furthermore, FSTLLM offers a flexible and extensible solution that allows existing state-of-the-art methods to be seamlessly integrated, resulting in performance improvements over their original implementations.

Our contributions are summarized as follows:

- **LLM Enhanced Graph Construction**: We propose a LLM-Enhanced Graph Construction module that utilizes LLMs to enhance graph construction by embedding contextual information of spatial nodes for time serie channels.

- **Domain Knowledge Injection**: We design a Domain Knowledge Injection module by fine-tuning an LLM to integrate domain-relevant knowledge with carefully crafted prompts. This design enables the model to account for human-like considerations of spatial and temporal characteristics, thereby delivering effective performance in forecasting.

- **Few-Shot Learning Integration**: We demonstrate that FSTLLM can augment various time series forecasting models, enhancing their performance in few-shot settings without updating their parameters.

- **Comprehensive Evaluation**: Experiments validate that FSTLLM outperforms baseline models on two real-world datasets under a limited data scenario by a large margin.

## 2. Related Work

### 2.1. Classical Neural Network-based Methods

Neural network-based approaches have been extensively applied to tackle time series forecasting tasks. Among them, LSTM (Hochreiter & Schmidhuber, 1997) and GRU (Chung et al., 2014) are two notable variants of Recurrent Neural Networks (RNNs) that are commonly used in time series forecasting (Bai et al., 2020; Shang et al., 2021; Shi et al., 2015). Recently, Transformer-based models (Zhou et al., 2021; Cirstea et al., 2022; Wu et al., 2021; Liu et al., 2022; Zhou et al., 2022; Zhang & Yan, 2023; Miao et al., 2024a; 2025) have shown demonstrated considerable promise in long-term time series forecasting, largely due to their capability to model long-range and pairwise temporal dependencies. For instance, Informer (Zhou et al., 2021) introduces a sparse self-attention mechanism, optimizing computational efficiency by utilizing Kullback-Leibler divergence for attention sparsity estimation. PatchTST (Nie et al., 2023) divides input sequences into patches as basic units, which are processed through a Transformer backbone to enhance

forecasting accuracy. In parallel, linear models such as DLinear (Liu et al., 2024c) have shown competitive performance by employing a decomposition strategy to separately model trend and residual components, offering a lightweight alternative to attention-based architectures. Despite these advances, both Transformer-based and linear models often neglect the spatial dependencies inherent in multivariate time series, leading to limited effectiveness in scenarios where spatial correlations among channels play a critical role.

## 2.2. Spatio-Temporal Graph Neural Networks

Spatio-Temporal Graph Neural Networks (STGNNs) have become as a fundamental framework for modeling spatio-temporal correlations in multivariate time series, particularly when the time series data exhibits explicit spatial structure. These models typically employ GNNs to capture spatial correlations via message passing across different time series channels (nodes), while temporal dependencies are modeled using established sequential models (e.g., RNNs and Transformers). By jointly learning spatial and temporal correlations, STGNNs have achieved strong performance across forecasting benchmarks in spatio-temporal data, such as in traffic flow and crime prediction. Early STGNN models such as DCRNN (Li et al., 2017) and T-GCN (Zhao et al., 2020) combine GRUs for temporal modeling with GNNs that operate on predefined graphs derived from prior knowledge, such as physical or geographical proximity. However, the use of static graphs can result in suboptimal performance when these predefined relationships fail to accurately reflect the true underlying spatial relationships among nodes. To address this limitation, recent models, such as AGCRN (Bai et al., 2020) and MTGNN (Wu et al., 2020), have incorporated the adaptive graph learning module into the STGNN framework. These models learn node embeddings, and dynamiclly compute the inner product of embeddings to construct adjacency matrices that represents spatial correlations for dependency modeling. Further advancements, including GTS (Shang et al., 2021) and STEP (Shao et al., 2022a), incorporate explicit graph learning mechanisms to capture complex, non-linear spatial dependencies using dedicated graph construction components. However, the performance of these models often come with increased data requirements and computational complexity, making them less effective in scenarios with limited training data.

## 2.3. Large Language Models

Recent advancements (Jin et al., 2024; Zhou et al., 2023; Chang et al., 2023; Pan et al., 2024; Liu et al., 2024a;b) in large language models (LLMs) have demonstrated promising potential in extending their applications to time series analysis, potentially in few-shot time series forecasting. GPT4TS (Zhou et al., 2023) adapts GPT2 for time series

tasks with only numerical temporal inputs, showing moderate performance across a range of time series analytical tasks. LLM4TS (Chang et al., 2023) introduces a two-stage fine-tuning approach for time series forecasting: the first stage involves task-specific pre-training to align the LLM with the structural characteristics of time series data, followed by a fine-tuning stage that leverages the model's forecasting capabilities. Time-LLM (Jin et al., 2024) presents an approach that reprograms numerical sequences of time series into contextual embedding, and employs the Prompt-as-Prefix method to generate predictions with LLM backbone. AutoTimes (Liu et al., 2024e) repurposes decoder-only LLMs for autoregressive time series forecasting by mapping time series inputs into the embedding space of language tokens. It enables the derivation of variable-length future predictions without updating the LLM weights. Time-MoE (Shi et al., 2025) introduces a scalable architecture based on a sparse mixture-of-experts paradigm, trained on the expansive datasets. With large-scale parameters, Time-MoE achieves strong predictive performance but exhibits diminished effectiveness in few-shot scenarios due to limitations in its expert routing mechanism. News2Forecast (Wang et al., 2024) enhances the performance by integrating social event signals through LLM-based agents using reflection and reasoning. It fine-tunes a pre-trained LLM to align textual and numerical data, thereby improving forecasting accuracy. TimeKD (Liu et al., 2025) combines calibrated LLMs with privileged knowledge distillation, utilizing a cross-modality teacher trained on both historical inputs and future ground-truth prompts, and a lightweight student model that is accessible to only historical inputs.

Despite their promising contributions, these methods disregard the spatial correlations among the channels of time series, which are critical for accurate multivariate forecasting in real-world spatio-temporal contexts. Besides, the internal reasoning capabilities of LLMs remain underutilized. Most existing approaches either fine-tune the LLM backbone directly on normalized temporal inputs or augment these inputs with basic task-specific prompts, thus significantly restricting both the reasoning capacity and overall predictive performance of the models.

## 3. Method

FSTLLM is an LLM-enhanced framework designed for few-shot time series forecasting. It advances forecasting performance across both spatial and temporal dimensions by leveraging the rich real-world knowledge encoded in LLMs. FSTLLM consists of three key components: LLM-Enhanced Graph Construction module, STGNN backbone module, and Domain Knowledge Injection module. The overall framework of FSTLLM is illustrated in Figure 1.

First, LLM-Enhanced Graph Construction module leverages

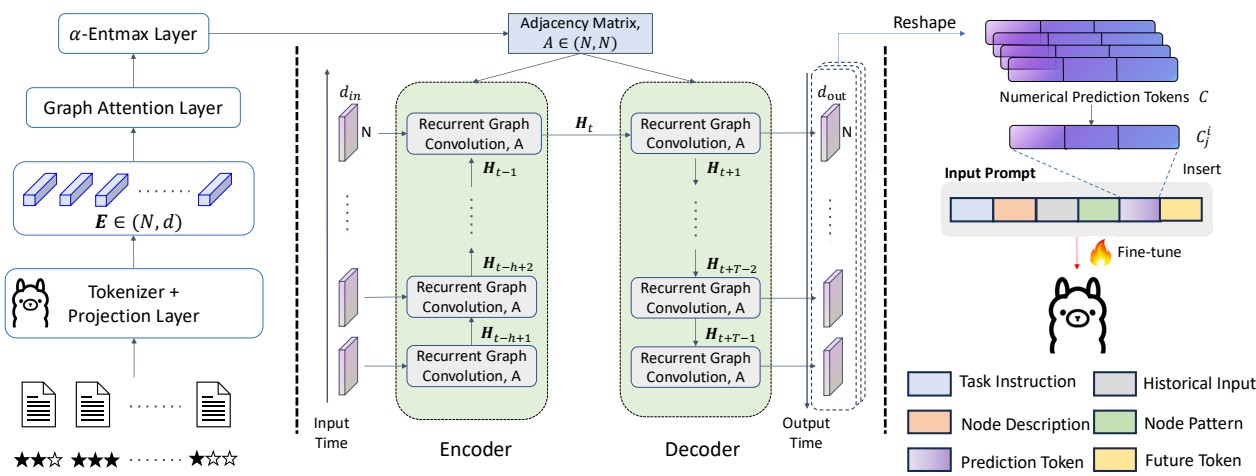

*Figure 1.* The overall architecture of the proposed FSTLLM framework. The LLM Enhanced Graph Construction Block is on the left, the STGNN module is in the middle, and the Domain Knowledge Injection module is on the right.

the contextual information encoded by LLMs, such as entity descriptions and user reviews, to construct an adjacency matrix **A** representing the spatial correlations. By incorporating semantically rich knowledge inferred by the LLM, the resulting adjacency matrix captures meaningful relationships between nodes that go beyond purely data-driven proximity. This context-aware graph construction is particularly advantageous in data-scarce scenarios, where conventional STGNNs may produce unstable or unreliable node embeddings due to insufficient training signals. Next, the generated adjacency matrix is fed into a STGNN model, which performs joint spatio-temporal modeling and outputs numerical prediction tokens. While this module is compatible with a range of existing STGNN architectures, its predictive accuracy may remain less effective with limited data availability in few-shot settings. To enhance the performance, relying on the reasoning capabilities of LLMs, Domain Knowledge Injection module is proposed to fine-tune an LLM to using a carefully curated, supervised dataset composed of prompts enriched with domain-specific knowledge to calibrate the numerical tokens predicted from STGNN. Unlike prior methods such as Time-LLM, which adopt generic task descriptions as prompts, our approach integrates node-specific descriptions, temporal pattern insights, and domain expertise into the prompt design, leading to more human-like contextual reasoning for LLMs. By seamlessly integrating these three modules, FSTLLM achieves robust and accurate few-shot time series forecasting, while advancing the capacity for spatio-temporal reasoning with real-world contextual awareness.

### 3.1. LLM Enhanced Graph Construction Module

In this study, we collect and extract a set of textual documents containing real-world contextual information rele-

vant to each time series node. Effectively leveraging node-specific content is critical for capturing the intricate spatio-temporal dependencies required for accurate forecasting. To this end, we employ a pre-trained LLM to encode node-specific contextual documents and generate semantically rich embeddings tailored to each node.

For example, in the case of a parking lot occupancy prediction task, node-specific documents include features such as parking rates, location details, maximum capacity, and user reviews. These textual inputs are processed using a LLaMA-2-7B model (Touvron et al., 2023), and the final-layer hidden state representations from the model are extracted as the node-specific initial embeddings, denoted as $\mathbf{H}_D \in \mathbb{R}^{N \times D}$ where $N$ represents the total number of nodes in the graph (e.g., the number of parking lots), and $D$ corresponds to the LLM dimension. To transform these representations into a form suitable for subsequent graph-based processing, we apply a Feed Forward Network ($\mathrm{FFN}_1$) to project $\mathbf{H}_D$ into a lower dimension of $d$. The resulting matrix, denoted by $\mathbf{E} \in \mathbb{R}^{N \times d}$, contains refined node embeddings, where each row $i$-th row $\mathbf{E}_i \in \mathbb{R}^d$ corresponds to the embedding of the $i$-th node. The embeddings $\mathbf{E}$ serve as inputs to the graph attention layer, which is formally defined as:

$$\bar{\mathbf{E}}_i = \oplus \left( \mathrm{repeat}(\mathbf{E}_i, N), \mathbf{E} \right) \qquad \in \mathbb{R}^{N \times 2d} \qquad (1)$$

$$\mathbf{Y}_i = \mathrm{FFN}_2(\bar{\mathbf{E}}_i) \qquad \in \mathbb{R}^{N \times 2} \qquad (2)$$

$$\mathbf{Y} = \mathrm{stack}([\mathbf{Y}_1, \mathbf{Y}_2, \dots, \mathbf{Y}_N]) \in \mathbb{R}^{N^2 \times 2} \qquad (3)$$

The operation $\mathrm{repeat}()$ duplicates the embedding $\mathbf{E}_i$ across $N$ rows, which is then concatenated with all node embeddings in $\mathbf{E}$ using the $\oplus$ operator, yielding a matrix $\bar{\mathbf{E}}_i \in \mathbb{R}^{N \times 2d}$. Each row of $\bar{\mathbf{E}}_i$ represents a pairwise combination between node $i$ and a candidate node. This matrix

serves as input to the Feed Forward Network (FFN₂), which produces $\mathbf{Y}_i \in \mathbb{R}^{N \times 2}$. Each row of $\mathbf{Y}_i$ denotes the pairwise relationship indicating the strength and the non-existent correlation between the node pair respectively.

By stacking the outputs $\mathbf{Y}_i$ for all nodes, we construct a matrix $\mathbf{Y} \in \mathbb{R}^{N^2 \times 2}$, where each row encodes the pairwise spatial correlation between nodes. To normalize these scores and regulate the sparsity of the resulting attention distribution, we apply the $\alpha$-Entmax function (Peters et al., 2019; Jiang et al., 2024), which generalizes softmax and enables sparse probability assignments. Subsequently, we use a another feed-forward layer $FFN_3 \in \mathbb{R}^{2 \times 1}$ to produce the final adjacency matrix $\mathbf{A} \in \mathbb{R}^{N \times N}$ for spatio-temporal modeling. The process is formally defined as:

$$\mathbf{A} = \text{FFN}_3(\alpha\text{-Entmax}(\mathbf{Y})) \tag{4}$$

### 3.2. STGNN Module

We integrate the obtained adjacency matrix $\mathbf{A}$ into a STGNN backbone to derive numerical prediction tokens, which are subsequently used in the prompt to fine-tune the LLM. The framework is flexible and supports the integration of any standard STGNN framework. In this work, we incorporate graph diffusion convolution into the classical GRU architecture described in GTS (Shang et al., 2021). Specifically, the standard matrix multiplication in the GRU is strategically replaced with a graph diffusion convolution operation, as defined below:

$$
\begin{aligned}
\mathbf{R}_t &= \sigma(\mathbf{W}_r \star_{\mathbf{A}} \oplus(\mathbf{X}_t, \mathbf{H}_{t-1}) + \mathbf{b}_r) \\
\mathbf{Z}_t &= \sigma(\mathbf{W}_z \star_{\mathbf{A}} \oplus(\mathbf{X}_t, \mathbf{H}_{t-1}) + \mathbf{b}_z) \\
\tilde{\mathbf{H}}_t &= \tanh(\mathbf{W}_h \star_{\mathbf{A}} \oplus (\mathbf{X}_t, \mathbf{R}_t \odot \mathbf{H}_{t-1}) + \mathbf{b}_h)) \\
\mathbf{H}_t &= \mathbf{Z}_t \odot \mathbf{H}_{t-1} + (1 - \mathbf{Z}_t) \odot \tilde{\mathbf{H}}_t
\end{aligned}
\tag{5}
$$

where $\mathbf{X}_t$ and $\mathbf{H}_t$ represent the input and hidden state at time step $t$, respectively. $\mathbf{R}_t$ and $\mathbf{Z}_t$ denote the reset gate and update gate at time step $t$. The sigmoid activation function is denoted by $\sigma$. $\mathbf{W}_r$, $\mathbf{W}_z$, and $\mathbf{W}_h$ are set of learnable model parameters used in the graph diffusion convolution operation. The graph convolution operation in Equation 5 is defined as follows:

$$\mathbf{W} \star_{\mathbf{A}} \mathbf{X} = \sum_{s=0}^{S-1} \mathbf{W}_s (\mathbf{D} + \mathbf{I})^{-1} \mathbf{A}^s \mathbf{X} \tag{6}$$

where $\mathbf{W} = \{\mathbf{W}_s\}_{s=1}^{S}$ represents the set of learnable parameters, $\mathbf{D} \in \mathbb{R}^{N \times N}$ is the degree matrix, and $S$ denotes the number of steps in the graph diffusion process. $\mathbf{X}$ represents the input to the graph diffusion convolution. The prediction $\hat{\mathbf{X}}_t$ at time step $t$ is obtained through a linear

transformation parameterized by the weight matrix $\mathbf{W}_x$ as shown in Equation 7.

$$\hat{\mathbf{X}}_t = \mathbf{H}_t \mathbf{W}_x \tag{7}$$

All learnable parameters are updated based on the prediction loss computed over the training datasets. For this purpose, we employ the Mean Absolute Error (MAE) as the loss:

$$\mathcal{L}(\mathbf{X}, \hat{\mathbf{X}}; \mathbf{\Theta}) = \frac{1}{NT} \sum_{t=t_0}^{t_0+T} \left\| \mathbf{X}_t - \hat{\mathbf{X}}_t \right\| \tag{8}$$

We aggregate all prediction results of $\hat{\mathbf{X}}$ across all horizons and nodes from the training/testing dataset, represented as $\mathbf{C} \in \mathbb{R}^{M \times N \times T}$. The tensor is then reshaped into numerical prediction tokens as $\mathbf{C} \in \mathbb{R}^{N \times M \times T}$, where $M$ denotes the number of training/testing samples, $N$ represents the number of nodes, and $T$ corresponds to the prediction horizons. These numerical tokens are subsequently utilized in the Domain Knowledge Injection module to fine-tune the LLM, enriching their predictive capabilities.

### 3.3. Domain Knowledge Injection Module

The Domain Knowledge Injection module is designed to enhance the model's ability to perform human-like reasoning over spatial and temporal contexts, thereby enabling superior few-shot forecasting performance. This is achieved by fine-tuning a LLM using carefully crafted prompts that integrate domain-specific knowledge along with the numerical prediction tokens $\mathbf{C}$. Specifically, we employ the supervised fine-tuning (SFT) on the LLaMA-2-7B (Touvron et al., 2023) model using QLoRA (Xu et al., 2024) to reduce the computational cost. Each prompt in the training dataset consists of six input components, as illustrated in Figure 1, with an detailed example provided in the Appendix A. The prompt is defined as follows:

- **Task Instruction:** Detailed instructions describing the specific forecasting task, including the domain of the time series, time-frequency, historical steps, and forecasting steps.

- **Node Description:** Node-specific descriptions are constructed by feeding the $j$-th node's documents and user reviews into the LLMs, which summarize the relevant details for the $j$-th node.

- **Node Pattern:** The $j$-th node's raw and limited training data is fed into the LLMs, which summarize patterns such as daily and weekly trends, peak and off-peak periods, and corresponding values.

- **Historical Input:** $\mathbf{X}_j^i \in \mathbb{R}^T$ represents the input time series for the $j$-th node used by the STGNN backbone.

Table 1. Performance comparison on Nottingham dataset.

| | 15 mins | | | 30 mins | | | 45 mins | | | 60 mins | | |
|---|---|---|---|---|---|---|---|---|---|---|---|---|
| | MAE | RMSE | MAPE | MAE | RMSE | MAPE | MAE | RMSE | MAPE | MAE | RMSE | MAPE |
| ARIMA | 12.98 | 41.73 | 8.69% | 21.16 | 63.63 | 14.98% | 24.71 | 77.56 | 18.91% | 28.49 | 82.47 | 23.60% |
| VAR | 11.65 | **41.44** | 8.30% | 27.70 | 67.06 | 16.88% | 30.06 | 77.17 | 31.52% | 32.19 | 91.85 | 49.00% |
| DCRNN | 7.93 | 44.64 | 5.65% | 13.21 | 61.94 | 9.76% | 18.37 | 76.80 | 13.53% | 24.10 | 89.67 | 20.68% |
| GraphWaveNet | 8.03 | 43.23 | 6.94% | 13.07 | 61.60 | 10.30% | 18.02 | 76.18 | 14.57% | 23.05 | 88.71 | 19.89% |
| STSGCN | 10.45 | 51.94 | 8.25% | 15.28 | 67.03 | 12.65% | 19.78 | 78.21 | 18.37% | 24.12 | 87.59 | 23.78% |
| GMAN | 42.09 | 102.41 | 29.01% | 43.87 | 106.62 | 30.86% | 45.53 | 110.03 | 33.11% | 47.13 | 113.23 | 35.74% |
| GTS | 7.89 | 42.79 | 5.54% | 12.72 | 60.75 | 9.50% | 17.80 | 75.08 | 13.51% | 23.06 | 87.31 | 20.40% |
| PatchTST | 12.54 | 48.42 | 12.93% | 18.78 | 66.65 | 19.08% | 25.03 | 81.57 | 24.06% | 31.25 | 94.62 | 29.69% |
| iTransformer | 7.91 | 44.03 | 6.83% | 13.12 | 63.43 | 10.33% | 19.10 | 77.79 | 14.77% | 26.09 | 91.80 | 22.66% |
| DLinear | 7.98 | 46.28 | 6.80% | 14.60 | 62.08 | 12.58% | 21.10 | 77.02 | 16.97% | 28.17 | 89.96 | 23.61% |
| GPT4TS | 8.02 | 44.09 | 7.06% | 13.16 | 62.51 | 10.79% | 18.61 | 77.23 | 15.78% | 24.24 | 90.00 | 21.55% |
| TimeLLM | 10.55 | 47.49 | 8.45% | 15.83 | 64.77 | 12.09% | 22.06 | 79.48 | 16.99% | 28.57 | 92.91 | 25.57% |
| FSTLLM | **7.83** | 47.93 | 6.59% | **11.53** | **60.11** | **9.47%** | **15.31** | **71.59** | **12.31%** | **19.09** | **80.90** | **17.76%** |
| | 75 mins | | | 90 mins | | | 105 mins | | | 120 mins | | |
| | MAE | RMSE | MAPE | MAE | RMSE | MAPE | MAE | RMSE | MAPE | MAE | RMSE | MAPE |
| ARIMA | 33.95 | 91.98 | 30.67% | 42.19 | 100.91 | 37.42% | 51.34 | 123.13 | 53.74% | 58.51 | 133.27 | 77.07% |
| VAR | 37.72 | 91.36 | 39.51% | 43.48 | 133.10 | 79.16% | 48.73 | 143.58 | 64.04% | 54.70 | 164.36 | 71.75% |
| DCRNN | 30.02 | 101.37 | 26.64% | 36.01 | 112.06 | 34.30% | 41.99 | 122.06 | 41.96% | 47.88 | 131.27 | 50.62% |
| GraphWaveNet | 28.52 | 100.33 | 24.16% | 34.12 | 110.26 | 33.22% | 39.59 | 120.34 | 37.51% | 45.16 | 128.57 | 47.50% |
| STSGCN | 28.49 | 95.53 | 30.55% | 32.69 | 102.47 | 37.57% | 36.42 | 108.72 | 44.28% | 39.87 | 114.03 | 52.04% |
| GMAN | 48.65 | 116.01 | 38.06% | 50.13 | 118.41 | 40.64% | 51.79 | 121.29 | 43.55% | 53.54 | 124.20 | 46.00% |
| GTS | 28.44 | 98.38 | 25.78% | 33.87 | 108.43 | 33.17% | 39.26 | 117.75 | 40.52% | 44.45 | 126.22 | 49.03% |
| PatchTST | 37.43 | 106.52 | 36.69% | 43.54 | 117.45 | 44.59% | 49.52 | 127.56 | 53.46% | 55.37 | 136.90 | 63.83% |
| iTransformer | 31.39 | 102.89 | 28.82% | 37.43 | 113.68 | 36.24% | 43.32 | 123.99 | 43.25% | 49.28 | 132.99 | 53.94% |
| DLinear | 34.87 | 101.61 | 31.10% | 42.09 | 112.45 | 37.87% | 49.20 | 122.25 | 44.07% | 56.17 | 131.31 | 52.37% |
| GPT4TS | 30.20 | 101.68 | 29.05% | 36.21 | 112.42 | 36.88% | 42.20 | 122.36 | 42.54% | 48.11 | 131.58 | 52.24% |
| TimeLLM | 34.50 | 104.53 | 30.78% | 40.18 | 115.20 | 38.08% | 46.47 | 125.55 | 46.66% | 53.58 | 136.11 | 58.96% |
| FSTLLM | **23.18** | **89.67** | **21.80%** | **26.65** | **95.82** | **26.51%** | **30.43** | **102.88** | **32.38%** | **33.89** | **109.54** | **39.09%** |

- **Numerical Prediction Token:** The numerical prediction $\mathbf{C}_j^i \in \mathbb{R}^T$ represents the possible future time series values predicted by the STGNN backbone. Specifically, the prediction token $\mathbf{C}_j^i$ represents the computed prediction token for the $j$-th node from the $i$-th training/testing sample, where $T$ denotes the prediction horizon.

- **Future Token:** $\mathbf{X}_j^{i+T} \in \mathbb{R}^T$ serves as the ground truth token corresponding to the future time steps for the $j$-th node during fine-tuning. At the evaluation stage, the future token is omitted.

During inference, the fine-tuned FSTLLM outputs context-aware predictions by integrating real-world contextual information, node-specific analyses (detailed in Appendix D), historical input time series, and numerical predictions derived from the STGNN backbone. FSTLLM jointly considers domain knowledge, temporal dynamics, and spatial correlations for accurate few-shot time series forecasting.

## 4. Experiments

In this section, we evaluate the effectiveness of the proposed FSTLLM method using two real-world datasets. We begin with a detailed description of the experimental setup,

followed by a comprehensive comparison of FSTLLM's performance against baseline methods to highlight its competitive advantages. To further substantiate the efficacy of FSTLLM, we conduct targeted experiments focusing on its core components, demonstrating their pivotal role in enhancing time series forecasting accuracy. Additionally, we perform experiments to demonstrate that our FSTLLM method can be adapted to other baseline methods to enhance their forecasting performance.

### 4.1. Experimental Setup

**Datasets**. We conduct experiments on two real-world time series datasets, with their detailed statistics provided in the Table 2. The Nottingham dataset contains parking lot availability data from 19 car parks in Nottingham, recorded at 15-minute intervals. We crawled this dataset from the official TramLink Nottingham website [1]. The ECL dataset is a subset of the Electricity dataset (Li et al., 2019), comprising hourly electricity consumption (measured in kilowatt-hours) from 19 clients. Both datasets are partitioned into training, validation, and testing sets with a split ratio of 70%/10%/20%. To create a few-shot experimental setup, the most recent week of data within the training set is set

[1]https://www.thetram.net/park-and-ride

*Table 2.* Statistics of our datasets

| Datasets | Time Frequency | Time range | Total Time Points |
|---|---|---|---|
| Nottingham | 15 mins | 26 Oct 2016 - 16 Feb 2017 | 10828 |
| ECL | 1 hour | 1 Jul 2016 - 2 Jul 2019 | 26304 |

aside.

**Baselines**. We have selected 12 time-series forecasting methods as comparison, including ARIMA (Williams & Hoel, 2003), VAR (Zivot & Wang, 2006), GTS (Shang et al., 2021), GraphWaveNet (Wu et al., 2019), STSGCN (Song et al., 2020), GMAN (Zheng et al., 2020), DCRNN (Li et al., 2017), DLinear (Liu et al., 2024c), PatchTST (Nie et al., 2023), GPT4TS (Zhou et al., 2023), iTransformer (Liu et al., 2024d), and TimeLLM (Jin et al., 2024). We also report other classical baselines such as AGCRN (Bai et al., 2020) and MTGNN (Wu et al., 2020) that fail to converge due to a lack of training data in the Appendix I.

**Metrics**. To evaluate the performance of different models, we adopt three widely-used metrics for multivariate time series forecasting, including Mean Absolute Error (MAE), Root Mean Squared Error (RMSE), and Mean Absolute Percentage Error (MAPE). A lower score indicates a better performance.

### 4.2. Performance of FSTLLM

Table 1 presents the forecasting performance across multiple horizons on the Nottingham dataset. The best result for each evaluation is highlighted in bold. The results demonstrate that the proposed FSTLLM model consistently outperforms other baseline methods across most scenarios. Specifically, FSTLLM achieves the best performance on 22 out of 24 evaluations, significantly surpassing state-of-the-art models such as GTS and VAR. Furthermore, FSTLLM exhibits consistently lower error rates and greater robustness across both short and long forecasting horizons, achieving a MAPE reduction of approximately 30% compared to other baselines. By efficiently modeling the spatio-temporal inherent in multivariate time series, FSTLLM surpasses both traditional and deep learning-based baselines, solidifying its position as a robust solution for few-shot time-series forecasting tasks.

Table 3 presents the forecasting performance across multiple horizons on the ECL dataset. The results demonstrate that the proposed FSTLLM model consistently outperforms other baseline methods across the majority of time horizons. Specifically, FSTLLM achieves the best performance in 25 out of 36 evaluations and the best two performances in 32 out of 36 evaluations, highlighting its robustness and superior accuracy compared to other state-of-the-art methods. Notably, FSTLLM exhibits significant improvements in MAPE, achieving a relative error reduction of over 50% compared to baseline methods such as GPT4TS and iTrans-

former. While attention-based models like iTransformer demonstrate moderate performance at extended time horizons due to their ability to capture distant temporal dependencies, they fall short of FSTLLM on short and mid-range horizons, primarily due to their reliance on large datasets and vulnerability to data sparsity. These results underscore the effectiveness of FSTLLM in accurately capturing both spatial and temporal correlations inherent in the data while maintaining robustness across various forecasting windows.

Appendix F presents the detailed comparison of average MAE, RMSE, and MAPE across all forecasting horizons on both datasets, further validating the superiority of the proposed FSTLLM model over existing baseline methods. The results reveal that FSTLLM achieves the lowest average errors across all three metrics, significantly outperforming all baselines. Overall, FSTLLM's superior performance positions it as a highly reliable solution for real-world time-series forecasting tasks, particularly in data-constrained environments.

**Reasoning.** FSTLLM exhibits superior reasoning ability by integrating real-world constraints and contextual information into its predictions. Unlike black-box models, which often provide numerical outputs without explanation, FSTLLM generates predictions that are both interpretable and grounded in domain knowledge. As shown in Appendix B, when forecasting parking lot availability, FSTLLM adjusts its prediction tokens to align with real-world constraints. Specifically, in a weekday scenario where parking demand decreases after a peak period (10 AM to 3 PM), FSTLLM adjusts predictions to reflect an expected rise in availability while adhering to the lot's capacity limit at the maximum capacity of 512 spaces. This adjustment reflects the model's capacity to incorporate real-world limits into its reasoning, ensuring predictions remain realistic and actionable for users.

By incorporating these human-like considerations and domain-specific knowledge, FSTLLM not only improves the accuracy of predictions but also enhances their utility to users. Users gain insights into the reasoning behind the predictions, allowing them to trust and act upon the results more confidently. This capability positions FSTLLM as a critical advancement over traditional deep learning methods, which lack this interpretability and reasoning depth.

*Table 3.* Performance comparison on ECL dataset.

| | 1 hour | | | 2 hours | | | 3 hours | | | 4 hours | | |
|---|---|---|---|---|---|---|---|---|---|---|---|---|
| | MAE | RMSE | MAPE | MAE | RMSE | MAPE | MAE | RMSE | MAPE | MAE | RMSE | MAPE |
| ARIMA | 2042 | 3774 | 17.64% | 2959 | 5407 | 19.98% | 4114 | 7044 | 32.78% | 4104 | 6907 | 33.50% |
| VAR | 1787 | 3682 | 15.40% | 2512 | 5206 | 19.71% | 2800 | 5271 | 26.26% | 3568 | 8206 | 30.21% |
| DCRNN | 2155 | 4694 | 19.05% | 2448 | 4839 | 23.34% | 2772 | 5110 | 27.92% | 2947 | 5223 | 31.61% |
| GraphWaveNet | **1497** | 3560 | **11.78%** | 2274 | 4659 | 17.91% | 2573 | 5120 | 22.22% | 2890 | 5575 | 26.81% |
| STSGCN | 3124 | 6719 | 22.15% | 3092 | 6385 | 20.65% | 3148 | 6328 | 21.07% | 3007 | 6013 | 21.44% |
| GMAN | 6181 | 10187 | 59.69% | 6249 | 10332 | 60.52% | 6313 | 10460 | 61.26% | 6364 | 10576 | 61.80% |
| GTS | 3517 | 6554 | 26.00% | 3676 | 6716 | 26.43% | 3827 | 6913 | 26.91% | 3915 | 6969 | 27.59% |
| PatchTST | 4789 | 8206 | 58.92% | 5526 | 9370 | 72.17% | 5955 | 10127 | 79.00% | 6204 | 10761 | 85.70% |
| iTransformer | 1764 | **2947** | 21.76% | 2289 | **4043** | 28.49% | 2874 | 5073 | 36.86% | 3399 | 5840 | 44.53% |
| DLinear | 2701 | 4409 | 28.87% | 3769 | 6023 | 38.86% | 4029 | 6533 | 44.50% | 4854 | 7790 | 54.45% |
| GPT4TS | 1672 | 3527 | 15.41% | 2384 | 4759 | 24.34% | 3009 | 5626 | 33.03% | 3621 | 6598 | 42.88% |
| TimeLLM | 4950 | 7640 | 61.74% | 5319 | 8308 | 69.56% | 5689 | 8722 | 75.36% | 5906 | 9072 | 79.59% |
| FSTLLM | 1663 | 3503 | 13.43% | **2264** | 4607 | **16.37%** | 2439 | 4938 | **16.65%** | 2520 | 5098 | **17.10%** |

| | 5 hours | | | 6 hours | | | 7 hours | | | 8 hours | | |
|---|---|---|---|---|---|---|---|---|---|---|---|---|
| | MAE | RMSE | MAPE | MAE | RMSE | MAPE | MAE | RMSE | MAPE | MAE | RMSE | MAPE |
| ARIMA | 4014 | 5912 | 31.62% | 4122 | 6858 | 40.10% | 4669 | 7054 | 43.48% | 5102 | 7773 | 48.64% |
| VAR | 3664 | 8582 | 35.44% | 4066 | 9677 | 40.96% | 4264 | 9812 | 47.27% | 5291 | 11933 | 63.51% |
| DCRNN | 3079 | 5298 | 33.82% | 3178 | 5369 | 35.24% | 3251 | 5446 | 36.14% | 3301 | 5537 | 36.61% |
| GraphWaveNet | 3153 | 5815 | 31.11% | 3274 | 5844 | 35.06% | 3364 | 5868 | 36.88% | 3340 | 5806 | 37.99% |
| STSGCN | 3059 | 6462 | 22.04% | 3061 | 6235 | 23.00% | 3331 | 7086 | 28.40% | 3041 | 6402 | 27.76% |
| GMAN | 6412 | 10665 | 62.39% | 6438 | 10721 | 62.53% | 6444 | 10753 | 62.39% | 6438 | 10761 | 62.25% |
| GTS | 3936 | 6938 | 27.70% | 3895 | 6838 | 27.50% | 3798 | 6694 | 27.22% | 3677 | 6561 | 26.75% |
| PatchTST | 6564 | 11094 | 89.05% | 6283 | 10977 | 87.91% | 5970 | 10491 | 81.81% | 5617 | 9721 | 74.23% |
| iTransformer | 3656 | 6333 | 49.80% | 3867 | 6536 | 53.33% | 3687 | 6308 | 50.24% | 3631 | 6246 | 48.18% |
| DLinear | 5978 | 9081 | 62.23% | 5242 | 8196 | 61.10% | 5259 | 8138 | 61.33% | 4989 | 7739 | 57.50% |
| GPT4TS | 4133 | 7277 | 53.54% | 4420 | 7514 | 60.10% | 4445 | 7437 | 60.31% | 4187 | 7099 | 55.27% |
| TimeLLM | 6112 | 9308 | 82.43% | 6121 | 9358 | 83.02% | 6118 | 9393 | 83.32% | 6071 | 9439 | 81.93% |
| FSTLLM | **2584** | **5201** | **17.56%** | **2648** | **5279** | **18.17%** | **2688** | **5333** | **18.33%** | **2760** | **5412** | **18.81%** |

| | 9 hours | | | 10 hours | | | 11 hours | | | 12 hours | | |
|---|---|---|---|---|---|---|---|---|---|---|---|---|
| | MAE | RMSE | MAPE | MAE | RMSE | MAPE | MAE | RMSE | MAPE | MAE | RMSE | MAPE |
| ARIMA | 4730 | 7792 | 46.96% | 5322 | 8499 | 51.73% | 5130 | 8304 | 47.66% | 5352 | 8455 | 48.49% |
| VAR | 5500 | 12319 | 67.96% | 5287 | 12864 | 65.76% | 5424 | 13536 | 64.49% | 6397 | 15368 | 58.73% |
| DCRNN | 3293 | 5618 | 36.24% | 3175 | 5544 | 34.27% | 2996 | 5410 | 31.12% | 2796 | 5314 | 27.59% |
| GraphWaveNet | 3267 | 5816 | 36.24% | 3090 | 5663 | 33.90% | 2895 | 5448 | 30.51% | 2652 | 5260 | 26.75% |
| STSGCN | 2966 | 5639 | 24.05% | 3136 | 7345 | 29.23% | 3099 | 7346 | 28.58% | 3205 | 7784 | 30.55% |
| GMAN | 6421 | 10742 | 61.58% | 6381 | 10681 | 60.66% | 6327 | 10607 | 59.67% | 6260 | 10499 | 58.79% |
| GTS | 3567 | 6484 | 26.07% | 3485 | 6476 | 25.28% | 3425 | 6528 | 24.53% | 3361 | 6606 | 23.80% |
| PatchTST | 5046 | 8623 | 63.58% | 4283 | 7351 | 51.63% | 3501 | 5953 | 40.11% | 2757 | 4537 | 28.15% |
| iTransformer | 3130 | **5362** | 37.98% | 2890 | **4750** | 34.24% | 2988 | **4360** | 29.50% | 1979 | **3562** | 19.79% |
| DLinear | 7298 | 11398 | 62.10% | 3947 | 6335 | 43.09% | 4830 | 7488 | 40.11% | 2803 | 4770 | 28.93% |
| GPT4TS | 3692 | 6365 | 44.80% | 3107 | 5802 | 34.54% | **2461** | 4578 | 25.08% | **1839** | 3597 | **17.00%** |
| TimeLLM | 5917 | 9060 | 78.78% | 5751 | 8727 | 73.46% | 5397 | 8204 | 67.24% | 4976 | 7613 | 60.40% |
| FSTLLM | **2822** | 5519 | **19.44%** | **2860** | 5460 | **19.80%** | **2894** | 5533 | **20.04%** | 2901 | 5478 | 20.37% |

## 4.3. Ablation Study

To further validate the contributions of the structured organization in FSTLLM, we conduct an ablation study by sequentially removing different modules of the framework. For this purpose, we define the following variants of FSTLLM:

(1) **FSTLLM-NoInjection:** This variant removes the Domain Knowledge Injection module entirely.

(1) **FSTLLM-NoLLM:** This variant replaces the LLM-Enhanced Graph Construction module with a commonly used cosine similarity-based graph learning operation, and removes the Domain Knowledge Injection module.

The performance of these variants is evaluated on the Nottingham dataset, and the results are summarized in Table 4. As shown, the removal of each component results in a noticeable decline in performance across all metrics. Among the variants, FSTLLM-NoInjection exhibits the most significant performance degradation, underscoring the importance of the Domain Knowledge Injection module. This result highlights the effectiveness of integrating real-world knowledge and human-like considerations into the forecasting framework, particularly in scenarios with limited training data.

*Table 4.* Ablation Study of FSTLLM on Nottingham dataset.

|  | MAE | MAPE | RMSE |
|---|---|---|---|
| FSTLLM-NoInjection | 25.1 | 23.9% | 94.5 |
| FSTLLM-NoLLM | 27.1 | 25.1% | 98.0 |
| FSTLLM | 21.0 | 20.7% | 82.3 |

*Table 5.* Few-Shot Learning Integration Performance on Nottingham dataset.

|  | MAE | MAPE | RMSE |
|---|---|---|---|
| GPT4TS | 27.6 | 27.0% | 92.7 |
| GPT4TS-FSTLLM | 21.8 | 21.8% | 84.3 |
| iTransformer | 28.4 | 26.8% | 96.7 |
| iTransformer-FSTLLM | 22.3 | 21.7% | 86.4 |

### 4.4. Few-Shot Learning Integration Study

In this section, we demonstrate that FSTLLM can augment existing state-of-the-art forecasting models, enhancing their performance in few-shot experimental setups without necessitating extensive modifications to their architectures and tremendous hyperparameter tuning. To evaluate this, we replace the numerical prediction tokens generated by the STGNN backbone with those produced by alternative transformer-based methods and assess the resulting performance improvements on the Nottingham dataset. To assess the plug-and-play potential of our LLM-enhanced framework, we substitute the numerical prediction tokens generated by the STGNN backbone with those from various transformer-based forecasting models. Specifically, we remove both the LLM-enhanced graph construction module and the STGNN backbone, and replace them with external transformer-based methods without modifying or retraining those models. This allows us to evaluate the generality of our integration strategy.

The results, summarized in Table 5, highlight significant performance gains for both GPT4TS and iTransformer models when integrated into the FSTLLM framework. Notably, the improvements are consistent across various metrics, underscoring the framework's ability to seamlessly adapt to different baseline models. These findings suggest that FSTLLM not only preserves its robust reasoning and forecasting performance but also amplifies the strengths of diverse forecasting architectures, making it a versatile and impactful enhancement for existing methods.

### 5. Conclusion

In this work, we presented FSTLLM, a novel framework that leverages LLMs to enhance few-shot time series forecasting by integrating domain knowledge and real-world constraints. The framework's LLM-Enhanced Graph Construction and Domain Knowledge Injection modules enable interpretable, accurate predictions by capturing contextual information and fine-tuning with numerical prediction tokens. Experimental results and ablation studies demonstrate FSTLLM's effectiveness in improving performance across various metrics, even with limited training data. Additionally, its adaptability to state-of-the-art models, such as GPT4TS and iTransformer, highlights its versatility without requiring significant architectural modifications. FSTLLM bridges the gap between deep learning-based forecasting

and human-like reasoning, offering a robust and adaptable solution for real-world applications.

### Acknowledgement

This study is supported by Singapore Telecommunications Limited (Singtel), through Singtel Cognitive and Artificial Intelligence Lab for Enterprises (SCALE@NTU). Xiucheng Li is supported by the National Natural Science Foundation of China under Grant No. 62206074, Guangdong Basic and Applied Basic Research Foundation under Grant No. 2025A1515012932, Shenzhen Science and Technology Program (JCYJ20241202123503005).

### Impact Statement

This work introduces FSTLLM, a novel framework that enhances time series forecasting by integrating large language models with spatio-temporal deep learning methods. By leveraging domain knowledge and real-world constraints, FSTLLM provides interpretable and accurate predictions, enabling better decision-making and resource allocation in scenarios with limited training data. There is no obvious negative impact on the community.

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

## A. Supervised Fine-tuning Prompt of FSTLLM.

In this section, we present a detailed prompt example used in supervised fine-tuning FSTLLM on Nottingham dataset as in Figure 2.

---

**FSTLLM Fine-tuning Prompt**

[INST] Role: You are an AI agent responsible for predicting the availability of parking lots. Objective: Your task is to forecast the number of available parking lots for the next 2 hours. To do this, you will analyze data from the past 2 hours of parking records, along with simulation-based predictions for the next 2 hours and other additional information that could affect the parking lots availability prediction values. Input Data: (1) Car park description: St Marks Place NCP Car Park - Affordable and Accessible Parking in Newark. The St Marks Place NCP Car Park, located on Lombard Street (NG24 1XT) in Newark and Sherwood, offers a spacious and affordable multi-story parking facility with 512 spaces, making it a convenient choice for visitors to the area. With a height restriction of 1.98 meters and dedicated disabled parking spaces, this car park caters to a variety of vehicles and accessibility needs. Open 24/7, St Marks Place provides flexible and budget-friendly pricing, with rates starting from £1.95 for 1 hour, £3.90 for 2 hours, and just £6.95 for a full 24-hour stay. Early bird discounts are available for those arriving before 9:00 am, making it an economical option for all-day parking. Payment options include ticketed parking, card payments, phone payments, and the NCP App (Location ID 32089), ensuring a convenient experience for all users. While the car park has earned a Parking Safety Award for its commitment to providing a secure environment, some users have noted that it can be tight to navigate, even for smaller cars, due to the high concrete sides. Despite this, St Marks Place NCP Car Park remains a practical and accessible choice for those looking for convenient parking in the Newark area. (2) Prediction horizon: you are given 2 hours historical input from 2017-01-24 14:31:14 to 2017-01-24 16:16:14 on Tuesday, and you are supposed to predict future parking lots from 2017-01-24 16:31:14 to 2017-01-24 18:16:14. (3) The Natural patterns of parking availability (e.g., peak and off-peak periods throughout the day) are as follow: Based on the given parking lot records and a detailed observation of the available parking spaces over the week, here is a more specific breakdown of when peaks (maximum availability) and dips (minimum availability) occur: On Weekdays (Monday to Friday): Peak availability: From 8 PM to 6 AM (reaching the maximum capacity of 512 spaces). This reflects the time when parking demand is lowest, typically during the late-night and early morning hours. Minimum availability: Generally occurs between 10 AM to 3 PM, with availability dropping to as low as 223–374 spaces. This indicates that these hours are the busiest, likely due to work or daily activities. On Weekends (Saturday and Sunday): Peak availability: Similar to weekdays, from 8 PM to 6 AM (maintaining the full capacity of 512 spaces), indicating lower demand during these hours. Minimum availability: Occurs between 11 AM to 4 PM, with availability dropping to as low as 248–374 spaces, showing moderate to high parking usage during midday hours. In summary, the parking lot consistently reaches full capacity during the late evening and early morning hours, with the highest usage occurring during the day, especially from mid-morning to mid-afternoon across both weekdays and weekends.. (4) Historical Records: Parking data for the past 2 hours [326. 339. 344. 348. 359. 369. 370. 382.]. (5) Simulation predictions: Forecasts for parking availability for the next 2 hours based on simulation models [389.95435 407.14072 426.157 443.97525 464.05444 484.45245 507.5884 524.72656]. Consider all provided information (historical records, simulation predictions, and additional factors) to predict parking lot availability for the next 2 hours with accuracy. Please analyze the data and provide only your final numerical predictions. DO NOT include any additional content in your answer. Strictly enclose your answer inside square brackets [] and provide exactly 8 numerical values. [/INST] [399. 425. 425. 451. 475. 499. 504. 512.].

| Task Instruction | Node Description | Node Pattern | Historical Input | Prediction Token | Future Token |

---

*Figure 2.* A sample of detailed FSTLLM supervised fine-tuning prompt.

## B. Evaluation Prompt of FSTLLM.

In this section, we present a detailed evaluation prompt example used by FSTLLM to provide predictions on Nottingham dataset as in Figure 3.

## C. Full Prompt of FSTLLM on ECL dataset.

In this section, we present the full prompt example used by FSTLLM on ECL dataset as in Figure 4.

## D. Node Description and Node Pattern.

In this work, we utilized ChatGPT-4o to assist in node description and node pattern generation. Specifically, we follow below template to generate node description and node pattern using ChatGPT-4o.

For Node Description Prompt:

"You are given a feature description from [carpark description link] and user reviews from [Google review link]. Please synthesize an inductive description of the carpark using content from both sources."

For Node Pattern Analysis Prompt:

"I will provide one week of parking lot records for a carpark in England, recorded every 15 minutes starting from 12:01 PM on 2016-10-26. Please describe the temporal usage patterns of the carpark, including variations throughout the day and across weekdays versus weekends. Indicate observed peak and low-demand periods, along with any consistent behavioral trends in parking availability. The records are: [extracted training data]."

---

**FSTLLM Evaluation Example**

[INST] Role: You are an AI agent responsible for predicting the availability of parking lots. Objective: Your task is to forecast the number of available parking lots for the next 2 hours. To do this, you will analyze data from the past 2 hours of parking records, along with simulation-based predictions for the next 2 hours and other additional information that could affect the parking lots availability prediction values. Input Data: (1) Car park description: St Marks Place NCP Car Park - Affordable and Accessible Parking in Newark. The St Marks Place NCP Car Park, located on Lombard Street (NG24 1XT) in Newark and Sherwood, offers a spacious and affordable multi-story parking facility with 512 spaces, making it a convenient choice for visitors to the area. With a height restriction of 1.98 meters and dedicated disabled parking spaces, this car park caters to a variety of vehicles and accessibility needs. Open 24/7, St Marks Place provides flexible and budget-friendly pricing, with rates starting from £1.95 for 1 hour, £3.90 for 2 hours, and just £6.95 for a full 24-hour stay. Early bird discounts are available for those arriving before 9:00 am, making it an economical option for all-day parking. Payment options include ticketed parking, card payments, phone payments, and the NCP App (Location ID 32089), ensuring a convenient experience for all users. While the car park has earned a Parking Safety Award for its commitment to providing a secure environment, some users have noted that it can be tight to navigate, even for smaller cars, due to the high concrete sides. Despite this, St Marks Place NCP Car Park remains a practical and accessible choice for those looking for convenient parking in the Newark area. (2) Prediction horizon: you are given 2 hours historical input from 2017-01-24 14:31:14 to 2017-01-24 16:16:14 on Tuesday, and you are supposed to predict future parking lots from 2017-01-24 16:31:14 to 2017-01-24 18:16:14. (3) The Natural patterns of parking availability (e.g., peak and off-peak periods throughout the day) are as follow: Based on the given parking lot records and a detailed observation of the available parking spaces over the week, here is a more specific breakdown of when peaks (maximum availability) and dips (minimum availability) occur: On Weekdays (Monday to Friday): Peak availability: From 8 PM to 6 AM (reaching the maximum capacity of 512 spaces). This reflects the time when parking demand is lowest, typically during the late-night and early morning hours. Minimum availability: Generally occurs between 10 AM to 3 PM, with availability dropping to as low as 223–374 spaces. This indicates that these hours are the busiest, likely due to work or daily activities. On Weekends (Saturday and Sunday): Peak availability: Similar to weekdays, from 8 PM to 6 AM (maintaining the full capacity of 512 spaces), indicating lower demand during these hours. Minimum availability: Occurs between 11 AM to 4 PM, with availability dropping to as low as 248–374 spaces, showing moderate to high parking usage during midday hours. In summary, the parking lot consistently reaches full capacity during the late evening and early morning hours, with the highest usage occurring during the day, especially from mid-morning to mid-afternoon across both weekdays and weekends.. (4) Historical Records: Parking data for the past 2 hours [326. 339. 344. 348. 359. 369. 370. 382.]. (5) Simulation predictions: Forecasts for parking availability for the next 2 hours based on simulation models [389.95435 407.14072 426.157 443.97525 464.05444 484.45245 507.5884 524.72656]. Consider all provided information (historical records, simulation predictions, and additional factors) to predict parking lot availability for the next 2 hours with accuracy. Please analyze the data and provide only your final numerical predictions. DO NOT include any additional content in your answer. Strictly enclose your answer inside square brackets [] and provide exactly 8 numerical values. [/INST] [398. 408. 433. 451. 466. 494. 512. 512.].\n</ref> To generate the predictions for the parking lot availability over the next two hours, I relied on a combination of the following factors: 1. Historical Records: The provided data showed an increasing trend over the past two hours, with values starting at 326 and ending at 382, indicating a gradual rise in the number of available parking spaces. 2. Simulation Predictions: The simulation-based predictions showed a continuous increase in availability, projecting values that ranged from 389 to 524. These values suggest an upward trend over the next two hours. 3. Peak and Off-Peak Patterns: The prediction horizon (from 16:31 to 18:16) falls just after the busiest period (10 AM to 3 PM) on a weekday, meaning parking availability is likely to rise as the day progresses. Based on the natural patterns, parking availability tends to increase after 3 PM and reaches its peak by 8 PM. 4. Capacity Constraints: The parking lot has a maximum capacity of 512 spaces, which acts as an upper bound for the prediction. This suggests that values nearing or above 512 should be adjusted to reflect the full occupancy of the car park. Based on these considerations, the predicted values were adjusted to align with the observed patterns and capacity limit, resulting in a final prediction of: [398, 408, 433, 451, 466, 494, 512, 512]. This reflects the expected gradual increase in availability, with the final values capped at the lot's maximum capacity.

| | Task Instruction | | Node Description | | Node Pattern | | Historical Input | | Prediction Token | | FSTLLM Answer |

*Figure 3.* A sample of detailed FSTLLM evaluation prompt.

## E. Implementation Details.

FSTLLM is fine-tuned on the training examples for 2 epochs on a Linux workstation with an Intel(R) Core(TM) i7-13700K CPU @ 5.40GHz and a NVIDIA A6000 GPU. We used 4-bit quantization (Liu et al., 2023) to obtain a more compact model representation, and low rank adaptation (LoRA) (Xu et al., 2024) to reduce the number of trainable parameters and decrease the GPU memory requirements. We set LoRA attention dimension to be 64 and an initial learning rate to be 2e-4 with Adam optimizer. We set $\alpha$ to be 2.0 in the $\alpha$-Entmax function and the depth of the graph diffusion convolution $S$ is set to 3. The hidden size of GRUs is set to 64. For all the baselines, we use their original implementations with minimum modifications required to run on our datasets. We set the sequence length and prediction horizon to 12 for the ECL dataset, as it exhibits notable daily and weekly patterns similar to traffic datasets such as METR-LA and PEMS-BAY, where a sequence length of 12 is commonly used. For the Nottingham dataset, since the average commuting time is less than 90 minutes[2], we set the sequence length and prediction horizon to 8 (equivalent to 120 minutes) to support parking lot predictions for users departing on their commutes.

Due to hardware limitations—using only a single GPU for inference—we could not employ data parallel methods to expedite inference, which are commonly used in other baseline methods such as (Jin et al., 2024; Zhou et al., 2023). Consequently, we extracted 19 users from the original 320 users in the electricity dataset to demonstrate the performance of FSTLLM and the baseline methods. However, we emphasize that FSTLLM is fully trainable on the entire electricity dataset with 320 nodes. The graph size of 320 nodes is manageable by the STGNN backbone (Shang et al., 2021; Shao et al., 2022a), and the Domain Knowledge Injection module fine-tunes LLaMA-2-7B in a node by node manner. Thus, the node count does not impact the computational cost during fine-tuning. We plan to evaluate FSTLLM and the baseline methods on the full electricity dataset in future work when we have access to multiple GPUs.

| **FSTLLM Fine-tuning Prompt_ECL** |
|---|
| \[INST] \[INST] Role: You are an AI agent responsible for predicting the user electricity consumption. Objective: Your task is to forecast the next 12 hour electricity consumption based on client past 12 hours historical records. To do this, you will analyse data from the past 12 hours records, along with simulation-based predictions for the next 12 hours and other additional information that could affect the electricity prediction values. Input Data: (1) Client description: The eletricity in the ECL dataset represents the hourly electricity consumption (Kwh) of 19 clients. This contains the electricity consumption of clients in certain region of 2 years. There are 20 clients recording every 1 hour intervals of electricity consumption. Each client has his own usage pattern and habbit, therefore, their behavior are different and should be consider seperately. You are now given the data from client no. 2, and you will be focusing on predicting future values for client no. 2.. (2) Prediction horizon: you are given 12 hour historical input from 2019-02-15 11:00:00 to 2019-02-15 22:00:00 on Friday, and you are supposed to predict future electricity consumption from 2019-02-15 23:00:00 to 2019-02-16 10:00:00. (3) The Natural patterns of electricity consumption of this client (e.g., peak and off-peak periods throughout the day) are as follow: This client electricity consumption over the week displays a clear pattern with higher usage during daytime hours and lower consumption at night, suggesting peak activity aligned with business hours. The highest consumption occurs on Thursday at 2 PM, reaching 99,167.0, likely indicating peak operational activity, while the lowest consumption is recorded on Sunday at 3 AM, at 30,043.0, reflecting minimal or no activity during late night hours. Weekdays generally show higher daytime usage, with consumption often surpassing 80,000.0 between late morning and early evening, which contrasts with the weekends, where usage is generally lower, rarely exceeding 75,000.0. This pattern suggests a drop in operational demand over the weekends, with energy needs substantially tapering off overnight and then gradually rising toward midday each day.. (4) Historical Records: electricity consumption for the past 12 hour [ 89501. 107127. 121709. 132627. 126668. 119752. 120418. 124918. 128501. 126043. 118501. 100376.]. (5) Simulation predictions: Forecasts for electricity consumption for the next 12 hour based on simulation models [78272.84 69957.67 56337.67 38943.453 28450.633 27399.629 27993.75 31317.69 46747.15 62374.055 71993.08 79677.27 ]. Consider all provided information (historical records, simulation predictions, and additional factors) to predict electricity consumption for the next 12 hour with accuracy. Please analyze the data and provide only your final numerical predictions. DO NOT include any additional content in your answer. Strictly enclose your answer inside square brackets [] and provide exactly 12 numerical values. [/INST] [91541. 88668. 79793. 59167. 54418. 54710. 53251. 51585. 61918. 69502. 72543 85376. ]. |
| **FSTLLM Evaluation Prompt_ECL** |
| \[INST] \[INST] Role: You are an AI agent responsible for predicting the user electricity consumption. Objective: Your task is to forecast the next 12 hour electricity consumption based on client past 12 hours historical records. To do this, you will analyse data from the past 12 hours records, along with simulation-based predictions for the next 12 hours and other additional information that could affect the electricity prediction values. Input Data: (1) Client description: The eletricity in the ECL dataset represents the hourly electricity consumption (Kwh) of 19 clients. This contains the electricity consumption of clients in certain region of 2 years. There are 20 clients recording every 1 hour intervals of electricity consumption. Each client has his own usage pattern and habbit, therefore, their behavior are different and should be consider seperately. You are now given the data from client no. 2, and you will be focusing on predicting future values for client no. 2.. (2) Prediction horizon: you are given 12 hour historical input from 2019-02-15 11:00:00 to 2019-02-15 22:00:00 on Friday, and you are supposed to predict future electricity consumption from 2019-02-15 23:00:00 to 2019-02-16 10:00:00. (3) The Natural patterns of electricity consumption of this client (e.g., peak and off-peak periods throughout the day) are as follow: This client electricity consumption over the week displays a clear pattern with higher usage during daytime hours and lower consumption at night, suggesting peak activity aligned with business hours. The highest consumption occurs on Thursday at 2 PM, reaching 99,167.0, likely indicating peak operational activity, while the lowest consumption is recorded on Sunday at 3 AM, at 30,043.0, reflecting minimal or no activity during late night hours. Weekdays generally show higher daytime usage, with consumption often surpassing 80,000.0 between late morning and early evening, which contrasts with the weekends, where usage is generally lower, rarely exceeding 75,000.0. This pattern suggests a drop in operational demand over the weekends, with energy needs substantially tapering off overnight and then gradually rising toward midday each day.. (4) Historical Records: electricity consumption for the past 12 hour [ 89501. 107127. 121709. 132627. 126668. 119752. 120418. 124918. 128501. 126043. 118501. 100376.]. (5) Simulation predictions: Forecasts for electricity consumption for the next 12 hour based on simulation models [78272.84 69957.67 56337.67 38943.453 28450.633 27399.629 27993.75 31317.69 46747.15 62374.055 71993.08 79677.27 ]. Consider all provided information (historical records, simulation predictions, and additional factors) to predict electricity consumption for the next 12 hour with accuracy. Please analyze the data and provide only your final numerical predictions. DO NOT include any additional content in your answer. Strictly enclose your answer inside square brackets [] and provide exactly 12 numerical values. [/INST] [74668. 67709. 50085. 36792. 36544. 36168. 35417. 48835. 63793. 74835. 88167.99 82334. ]. I hope this answers your question. Please let me know if you have any other questions or need further assistance. |

*Figure 4.* A sample of detailed FSTLLM fine-tuning and evaluation prompt on ECL dataset.

| Nottingham | FSTLLM | ARIMA | VAR | DCRNN | GraphWaveNet | STSGCN | GMAN | GTS | PatchTST | iTransformer | DLinear | GPT4TS | TimeLLM |
|---|---|---|---|---|---|---|---|---|---|---|---|---|---|
| MAE | 21.0 | 34.2/38.6% | 35.8/41.3% | 27.4/23.5% | 26.2/19.9% | 25.8/18.9% | 47.8/56.1% | 25.9/19.1% | 34.2/38.6% | 28.4/26.1% | 31.8/33.9% | 27.6/23.9% | 31.5/33.3% |
| RMSE | 82.3 | 89.3/7.9% | 101.2/18.7% | 92.5/11.0% | 91.2/9.8% | 88.2/6.7% | 114.0/27.8% | 89.6/8.1% | 97.5/15.6% | 96.7/14.9% | 92.9/11.4% | 92.7/11.3% | 95.8/14.1% |
| MAPE(%) | 20.7 | 33.1/37.4% | 45.0/53.9% | 25.4/18.3% | 28.9/28.4% | 28.4/27.2% | 37.1/44.1% | 24.7/16.0% | 35.5/41.6% | 26.8/22.6% | 28.2/26.4% | 27.0/23.2% | 29.7/30.2% |
| **ECL** | **FSTLLM** | **ARIMA** | **VAR** | **DCRNN** | **GraphWaveNet** | **STSGCN** | **GMAN** | **GTS** | **PatchTST** | **iTransformer** | **DLinear** | **GPT4TS** | **TimeLLM** |
| MAE | 2294 | 3447/33.4% | 2866/20.0% | 2680/14.4% | 2859/19.8% | 3106/26.1% | 6304/63.6% | 3774/39.2% | 5808/60.5% | 2984/23.1% | 4266/46.2% | 2964/22.6% | 5595/59.0% |
| RMSE | 4669 | 5809/19.6% | 6189/24.6% | 5033/7.2% | 5370/13.1% | 6645/29.7% | 10444/55.3% | 6818/31.5% | 9912/52.9% | 5113/8.7% | 6767/31.0% | 5557/16.0% | 8610/45.8% |
| MAPE(%) | 16.2 | 27.1/40.2% | 25.4/36.1% | 27.2/40.3% | 28.9/43.9% | 24.9/34.9% | 61.1/73.5% | 26.9/39.8% | 76.9/78.9% | 37.9/57.9% | 45.8/64.6% | 33.8/52.1% | 73.7/78.0% |

*Table 6.* Average MAE, RMSE, and MAPE for different methods, along with the percentage improvement of FSTLLM over each method (shown as Avg/Improvement %).

## F. Average MAE, RMSE, and MAPE Comparison.

## G. Data efficiency comparison.

To evaluate data efficiency, we compare FSTLLM trained on just 3 days of data with a representative subset of baselines trained on both 3 days of data and 30 days of data, using the Nottingham dataset. The results are summarized in Table 7. These results demonstrate that FSTLLM consistently outperforms strong baselines under limited data scenarios. This further validates the robustness and adaptability of our framework in few-shot settings. Furthermore, the results show that FSTLLM with only 3 days of training data outperforms all baselines trained with 10× more data, highlighting the strong data efficiency of our approach.

*Table 7.* Performance Comparison of on Different Training Days (3 Days and 30 Days)

| Method | MAE | RMSE | MAPE (%) |
|---|---|---|---|
| FSTLLM (3 days) | 22.84 | 83.68 | 22.33 |
| GTS (3 days) | 29.57 | 87.44 | 26.04 |
| GPT4TS (3 days) | 33.24 | 93.92 | 26.95 |
| PatchTST (3 days) | 34.65 | 97.45 | 32.38 |
| DLinear (3 days) | 37.52 | 95.87 | 30.62 |
| GTS (30 days) | 23.54 | 90.88 | 23.02 |
| GPT4TS (30 days) | 27.50 | 86.88 | 25.70 |
| PatchTST (30 days) | 33.07 | 95.20 | 30.13 |
| DLinear (30 days) | 32.82 | 92.64 | 26.63 |

## H. The $\alpha$-Entmax function.

In multivariate time series forecasting, the spatial correlations normalized using the Softmax function (Bai et al., 2020; Wu et al., 2020; Cirstea et al., 2021) often include a significant proportion of low-weight entries due to the nature of the Softmax function. Such low-weight entries generally imply little to no similarity in trend or seasonality. Applying graph convolution directly to these low-weight entries can lead to inaccurate message passing and dilute the focus on the node of interest, thereby diminishing the effectiveness of graph convolution. To address this issue, the $\alpha$-Entmax (Tezekbayev et al., 2021) is used, which incorporates a tunable hyperparameter $\alpha$. This approach mitigates the influence of distant nodes while amplifying the impact of closer nodes. The $\alpha$-Entmax function provides greater control over the normalized attention scores $\mathbf{Z}$, which represent spatial correlations. The $\alpha$-Entmax is defined as follows, where $[x]_+ := \max x, 0$:

$$\alpha\text{-Entmax}(\mathbf{z}) = [(\alpha - 1)\mathbf{z} - \tau\mathbf{1}]_+^{1/\alpha - 1} \tag{9}$$

Here, $\mathbf{z} \in \mathbb{R}^d$, and $\tau : \mathbb{R}^d \to \mathbb{R}$ is derived as:

$$\sum_j [(\alpha - 1)z_j - \tau(\mathbf{z})]_+^{\frac{1}{\alpha - 1}} = 1 \tag{10}$$

The $\alpha$-Entmax function generalizes the Softmax (with $\alpha = 1.0$) and Sparsemax (Martins & Astudillo, 2016) (with $\alpha = 2.0$), offering enhanced flexibility compared to the Softmax function. In our experiments, we employ an $\alpha$ value of 2.0 to suppress the information flow from the noise nodes.

## I. Full Forecasting Performance Comparison.

We present the full forecasting performance across multiple horizons on the Nottingham and the ECL datasets in this section, including AGCRN and MTGNN. The best result for each evaluation is highlighted in bold, and the second-best result for each evaluation is underlined.

---

[2]https://maps.dft.gov.uk/transport-statistics-finder/index.html

*Table 8.* Performance comparison on Nottingham dataset.

| | 15 mins | | | 30 mins | | | 45 mins | | | 60 mins | | |
|---|---|---|---|---|---|---|---|---|---|---|---|---|
| | MAE | RMSE | MAPE | MAE | RMSE | MAPE | MAE | RMSE | MAPE | MAE | RMSE | MAPE |
| ARIMA | 12.98 | 41.73 | 8.69% | 21.16 | 63.63 | 14.98% | 24.71 | 77.56 | 18.91% | 28.49 | 82.47 | 23.60% |
| VAR | 11.65 | **41.44** | 8.30% | 27.70 | 67.06 | 16.88% | 30.06 | 77.17 | 31.52% | 32.19 | 91.85 | 49.00% |
| DCRNN | 7.93 | 44.64 | 5.65% | 13.21 | 61.94 | 9.76% | 18.37 | 76.80 | 13.53% | 24.10 | 89.67 | 20.68% |
| GraphWaveNet | 8.03 | 43.23 | 6.94% | 13.07 | 61.60 | 10.30% | 18.02 | 76.18 | 14.57% | 23.05 | 88.71 | 19.89% |
| AGCRN | 153.09 | 282.06 | 32.03% | 151.12 | 281.27 | 30.72% | 149.72 | 280.77 | 30.0% | 149.37 | 280.89 | 30.88% |
| MTGNN | 7.94 | 42.92 | 6.76% | 234.26 | 314.58 | 281.20% | 234.25 | 314.59 | 281.09% | 234.25 | 314.62 | 281.00% |
| STSGCN | 10.45 | 51.94 | 8.25% | 15.28 | 67.03 | 12.65% | 19.78 | 78.21 | 18.37% | 24.12 | 87.59 | 23.78% |
| GMAN | 42.09 | 102.41 | 29.01% | 43.87 | 106.62 | 30.86% | 45.53 | 110.03 | 33.11% | 47.13 | 113.23 | 35.74% |
| GTS | 7.89 | 42.79 | **5.54%** | 12.72 | 60.75 | 9.50% | 17.80 | 75.08 | 13.51% | 23.06 | 87.31 | 20.40% |
| PatchTST | 12.54 | 48.42 | 12.93% | 18.78 | 66.65 | 19.08% | 25.03 | 81.57 | 24.06% | 31.25 | 94.62 | 29.69% |
| iTransformer | 7.91 | 44.03 | 6.83% | 13.12 | 63.43 | 10.33% | 19.10 | 77.79 | 14.77% | 26.09 | 91.80 | 22.66% |
| DLinear | 7.98 | 46.28 | 6.80% | 14.60 | 62.08 | 12.58% | 21.10 | 77.02 | 16.97% | 28.17 | 89.96 | 23.61% |
| GPT4TS | 8.02 | 44.09 | 7.06% | 13.16 | 62.51 | 10.79% | 18.61 | 77.23 | 15.78% | 24.24 | 90.00 | 21.55% |
| TimeLLM | 10.55 | 47.49 | 8.45% | 15.83 | 64.77 | 12.09% | 22.06 | 79.48 | 16.99% | 28.57 | 92.91 | 25.57% |
| FSTLLM | **7.83** | 47.93 | 6.59% | **11.53** | **60.11** | **9.47%** | **15.31** | **71.59** | **12.31%** | **19.09** | **80.90** | **17.76%** |
| | 75 mins | | | 90 mins | | | 105 mins | | | 120 mins | | |
| | MAE | RMSE | MAPE | MAE | RMSE | MAPE | MAE | RMSE | MAPE | MAE | RMSE | MAPE |
| ARIMA | 33.95 | 91.98 | 30.67% | 42.19 | 100.91 | 37.42% | 51.34 | 123.13 | 53.74% | 58.51 | 133.27 | 77.07% |
| VAR | 37.72 | 91.36 | 39.51% | 43.48 | 133.10 | 79.16% | 48.73 | 143.58 | 64.04% | 54.70 | 164.36 | 71.75% |
| DCRNN | 30.02 | 101.37 | 26.64% | 36.01 | 112.06 | 34.30% | 41.99 | 122.06 | 41.96% | 47.88 | 131.27 | 50.62% |
| GraphWaveNet | 28.52 | 100.33 | 24.16% | 34.12 | 110.26 | 33.22% | 39.59 | 120.34 | 37.51% | 45.16 | 128.57 | 47.50% |
| AGCRN | 150.31 | 281.51 | 32.29% | 152.16 | 282.45 | 35.29% | 154.89 | 283.94 | 38.89% | 157.49 | 285.15 | 43.43% |
| MTGNN | 234.24 | 314.64 | 280.92% | 234.24 | 314.66 | 280.85% | 234.25 | 314.69 | 280.79% | 234.26 | 314.72 | 280.75% |
| STSGCN | 28.49 | 95.53 | 30.55% | 32.69 | 102.47 | 37.57% | 36.42 | 108.72 | 44.28% | 39.87 | 114.03 | 52.04% |
| GMAN | 48.65 | 116.01 | 38.06% | 50.13 | 118.41 | 40.64% | 51.79 | 121.29 | 43.55% | 53.54 | 124.20 | 46.00% |
| GTS | 28.44 | 98.38 | 25.78% | 33.87 | 108.43 | 33.17% | 39.26 | 117.75 | 40.52% | 44.45 | 126.22 | 49.03% |
| PatchTST | 37.43 | 106.52 | 36.69% | 43.54 | 117.45 | 44.59% | 49.52 | 127.56 | 53.46% | 55.37 | 136.90 | 63.83% |
| iTransformer | 31.39 | 102.89 | 28.82% | 37.43 | 113.68 | 36.24% | 43.32 | 123.99 | 43.25% | 49.28 | 132.99 | 53.94% |
| DLinear | 34.87 | 101.61 | 31.10% | 42.09 | 112.45 | 37.87% | 49.20 | 122.25 | 44.07% | 56.17 | 131.31 | 52.37% |
| GPT4TS | 30.20 | 101.68 | 29.05% | 36.21 | 112.42 | 36.88% | 42.20 | 122.36 | 42.54% | 48.11 | 131.58 | 52.24% |
| TimeLLM | 34.50 | 104.53 | 30.78% | 40.18 | 115.20 | 38.08% | 46.47 | 125.55 | 46.66% | 53.58 | 136.11 | 58.96% |
| FSTLLM | **23.18** | **89.67** | **21.80%** | **26.65** | **95.82** | **26.51%** | **30.43** | **102.88** | **32.38%** | **33.89** | **109.54** | **39.09%** |

*Table 9.* Performance comparison on ECL dataset.

| | 1 hour | | | 2 hours | | | 3 hours | | | 4 hours | | |
|---|---|---|---|---|---|---|---|---|---|---|---|---|
| | MAE | RMSE | MAPE | MAE | RMSE | MAPE | MAE | RMSE | MAPE | MAE | RMSE | MAPE |
| ARIMA | 2042 | 3774 | 17.64% | 2959 | 5407 | 19.98% | 4114 | 7044 | 32.78% | 4104 | 6907 | 33.50% |
| VAR | 1787 | 3682 | 15.40% | 2512 | 5206 | 19.71% | 2800 | 5271 | 26.26% | 3568 | 8206 | 30.21% |
| DCRNN | 2155 | 4694 | 19.05% | 2448 | 4839 | 23.34% | 2772 | 5110 | 27.92% | 2947 | 5223 | 31.61% |
| GraphWaveNet | 1497 | 3560 | 11.78% | 2274 | 4659 | 17.91% | 2573 | 5120 | 22.22% | 2890 | 5575 | 26.81% |
| AGCRN | 16523 | 25513 | 98.80% | 16524 | 25514 | 98.80% | 16524 | 25514 | 98.79% | 16524 | 25514 | 98.80% |
| MTGNN | 1705 | 3578 | 15.23% | 11527 | 19531 | 141.79% | 11527 | 19531 | 141.79% | 11527 | 19532 | 141.79% |
| STSGCN | 3124 | 6719 | 22.15% | 3092 | 6385 | 20.65% | 3148 | 6328 | 21.07% | 3007 | 6013 | 21.44% |
| GMAN | 6181 | 10187 | 59.69% | 6249 | 10332 | 60.52% | 6313 | 10460 | 61.26% | 6364 | 10576 | 61.80% |
| GTS | 3517 | 6554 | 26.00% | 3676 | 6716 | 26.43% | 3827 | 6913 | 26.91% | 3915 | 6969 | 27.59% |
| PatchTST | 4789 | 8206 | 58.92% | 5526 | 9370 | 72.17% | 5955 | 10127 | 79.00% | 6204 | 10761 | 85.70% |
| iTransformer | 1764 | 2947 | 21.76% | 2289 | 4043 | 28.49% | 2874 | 5073 | 36.86% | 3399 | 5840 | 44.53% |
| DLinear | 2701 | 4409 | 28.87% | 3769 | 6023 | 38.86% | 4029 | 6533 | 44.50% | 4854 | 7790 | 54.45% |
| GPT4TS | 1672 | 3527 | 15.41% | 2384 | 4759 | 24.34% | 3009 | 5626 | 33.03% | 3621 | 6598 | 42.88% |
| TimeLLM | 4950 | 7640 | 61.74% | 5319 | 8308 | 69.56% | 5689 | 8722 | 75.36% | 5906 | 9072 | 79.59% |
| FSTLLM | 1663 | 3503 | 13.43% | 2264 | 4607 | 16.37% | 2439 | 4938 | 16.65% | 2520 | 5098 | 17.10% |
| | 5 hours | | | 6 hours | | | 7 hours | | | 8 hours | | |
| | MAE | RMSE | MAPE | MAE | RMSE | MAPE | MAE | RMSE | MAPE | MAE | RMSE | MAPE |
| ARIMA | 4014 | 5912 | 31.62% | 4122 | 6858 | 40.10% | 4669 | 7054 | 43.48% | 5102 | 7773 | 48.64% |
| VAR | 3664 | 8582 | 35.44% | 4066 | 9677 | 40.96% | 4264 | 9812 | 47.27% | 5291 | 11933 | 63.51% |
| DCRNN | 3079 | 5298 | 33.82% | 3178 | 5369 | 35.24% | 3251 | 5446 | 36.14% | 3301 | 5537 | 36.61% |
| GraphWaveNet | 3153 | 5815 | 31.11% | 3274 | 5844 | 35.06% | 3364 | 5868 | 36.88% | 3340 | 5806 | 37.99% |
| AGCRN | 16525 | 25515 | 98.81% | 16524 | 25514 | 98.81% | 16524 | 25514 | 98.79% | 16523 | 25513 | 98.80% |
| MTGNN | 11527 | 19532 | 141.79% | 11527 | 19532 | 141.79% | 11527 | 19532 | 141.79% | 11527 | 19532 | 141.79% |
| STSGCN | 3059 | 6462 | 22.04% | 3061 | 6235 | 23.00% | 3331 | 7086 | 28.40% | 3041 | 6402 | 27.76% |
| GMAN | 6412 | 10665 | 62.39% | 6438 | 10721 | 62.53% | 6444 | 10753 | 62.39% | 6438 | 10761 | 62.25% |
| GTS | 3936 | 6938 | 27.70% | 3895 | 6838 | 27.50% | 3798 | 6694 | 27.22% | 3677 | 6561 | 26.75% |
| PatchTST | 6564 | 11094 | 89.05% | 6283 | 10977 | 87.91% | 5970 | 10491 | 81.81% | 5617 | 9721 | 74.23% |
| iTransformer | 3656 | 6333 | 49.80% | 3867 | 6536 | 53.33% | 3687 | 6308 | 50.24% | 3631 | 6246 | 48.18% |
| DLinear | 5978 | 9081 | 62.23% | 5242 | 8196 | 61.10% | 5259 | 8138 | 61.33% | 4989 | 7739 | 57.50% |
| GPT4TS | 4133 | 7277 | 53.54% | 4420 | 7514 | 60.10% | 4445 | 7437 | 60.31% | 4187 | 7099 | 55.27% |
| TimeLLM | 6112 | 9308 | 82.43% | 6121 | 9358 | 83.02% | 6118 | 9393 | 83.32% | 6071 | 9439 | 81.93% |
| FSTLLM | 2584 | 5201 | 17.56% | 2648 | 5279 | 18.17% | 2688 | 5333 | 18.33% | 2760 | 5412 | 18.81% |
| | 9 hours | | | 10 hours | | | 11 hours | | | 12 hours | | |
| | MAE | RMSE | MAPE | MAE | RMSE | MAPE | MAE | RMSE | MAPE | MAE | RMSE | MAPE |
| ARIMA | 4730 | 7792 | 46.96% | 5322 | 8499 | 51.73% | 5130 | 8304 | 47.66% | 5352 | 8455 | 48.49% |
| VAR | 5500 | 12319 | 67.96% | 5287 | 12864 | 65.76% | 5424 | 13536 | 64.49% | 6397 | 15368 | 58.73% |
| DCRNN | 3293 | 5618 | 36.24% | 3175 | 5544 | 34.27% | 2996 | 5410 | 31.12% | 2796 | 5314 | 27.59% |
| GraphWaveNet | 3267 | 5816 | 36.24% | 3090 | 5663 | 33.90% | 2895 | 5448 | 30.51% | 2652 | 5260 | 26.75% |
| AGCRN | 16522 | 25512 | 98.81% | 16519 | 25510 | 98.80% | 16518 | 25508 | 98.79% | 16516 | 25506 | 98.80% |
| MTGNN | 11527 | 19531 | 141.81% | 11527 | 19530 | 141.84% | 11527 | 19530 | 141.88% | 11527 | 19529 | 141.91% |
| STSGCN | 2966 | 5639 | 24.05% | 3136 | 7345 | 29.23% | 3099 | 7346 | 28.58% | 3205 | 7784 | 30.55% |
| GMAN | 6421 | 10742 | 61.58% | 6381 | 10681 | 60.66% | 6327 | 10607 | 59.67% | 6260 | 10499 | 58.79% |
| GTS | 3567 | 6484 | 26.07% | 3485 | 6476 | 25.28% | 3425 | 6528 | 24.53% | 3361 | 6606 | 23.80% |
| PatchTST | 5046 | 8623 | 63.58% | 4283 | 7351 | 51.63% | 3501 | 5953 | 40.11% | 2757 | 4537 | 28.15% |
| iTransformer | 3130 | 5362 | 37.98% | 2890 | 4750 | 34.24% | 2988 | 4360 | 29.50% | 1979 | 3562 | 19.79% |
| DLinear | 7298 | 11398 | 62.10% | 3947 | 6335 | 43.09% | 4830 | 7488 | 40.11% | 2803 | 4770 | 28.93% |
| GPT4TS | 3692 | 6365 | 44.80% | 3107 | 5802 | 34.54% | 2461 | 4578 | 25.08% | 1839 | 3597 | 17.00% |
| TimeLLM | 5917 | 9060 | 78.78% | 5751 | 8727 | 73.46% | 5397 | 8204 | 67.24% | 4976 | 7613 | 60.40% |
| FSTLLM | 2822 | 5519 | 19.44% | 2860 | 5460 | 19.80% | 2894 | 5533 | 20.04% | 2901 | 5478 | 20.37% |

