# OpenReview forum: "FSTLLM: Spatio-Temporal LLM for Few Shot Time Series Forecasting"
_ICML.cc/2025/Conference — ICML 2025 poster_

### Official Review · Reviewer_KEzm · 2025-02-24

**Overall Recommendation:** 4

**Summary:**

The paper presents FSTLLM, a novel Spatio-Temporal Large Language Model (LLM) framework designed for few-shot time series forecasting. The model effectively integrates domain knowledge through a fine-tuned LLM and a graph-based learning approach to capture spatial-temporal correlations. Experimental results on real-world datasets demonstrate superior performance compared to existing baselines.

**Claims And Evidence:**

The claim that "The model effectively integrates domain knowledge through a fine-tuned LLM and a graph-based learning approach to capture spatio-temporal correlations" is well-supported by experimental results on real-world datasets. Additionally, the case study effectively demonstrates FSTLLM’s reasoning ability derived from domain knowledge.

**Essential References Not Discussed:**

The paper comprehensively covers relevant literature, but I suggest including the following recent work: ‘From News to Forecast: Integrating Event Analysis in LLM-Based Time Series Forecasting with Reflection’ (NeurIPS 2024). This study also fine-tunes an LLM in a textual format to address time-series forecasting challenges.

**Experimental Designs Or Analyses:**

The experimental design is sound. Specifically, FSTLLM significantly enhances few-shot forecasting performance of STGNNs while also incorporating reasoning ability, which is crucial for end users. However, some points need to be addressed:
1. The paper does not clearly explain how FSTLLM crafts its node descriptions and node pattern analyses, as introduced in Section 3.3 (Domain Knowledge Injection Module).
2. The study presents fine-tuning and inference case study results only on the Nottingham dataset. Including a demonstration on the ECL dataset would provide a more comprehensive understanding of the model's applicability.

**Methods And Evaluation Criteria:**

The proposed method primarily applies a fine-tuned LLM to enhance baseline methods such as STGNNs. The evaluation criteria and settings are consistent with existing works. Both the methodology and evaluation framework are appropriate for the problem at hand.

**Other Comments Or Suggestions:**

1. Page 1, Lines 11–14: "The fundamental of time series forecasting methodologies" → Should be "The fundamentals of time series forecasting methodologies".
2. Page 1, Lines 46–47: "Accurate forecasting requires precise modeling on two dimensions" → Should be "Accurate forecasting requires precise modeling of two dimensions".
3. Page 4, Lines 206–210: "these textual data" → Should be "this textual data".

**Other Strengths And Weaknesses:**

Strength:
1. The paper introduces FSTLLM, a novel framework that effectively integrates LLMs with Spatio-Temporal Graph Neural Networks (STGNNs) to enhance time-series forecasting.
2. Experimental results demonstrate that FSTLLM consistently outperforms state-of-the-art baselines across multiple forecasting horizons on real-world datasets.
3. The adaptable architecture of FSTLLM allows integration with existing time-series forecasting models.

Weakness:
1. As mentioned in the ‘Experimental Designs Or Analyses’ section, it is important to clarify how node descriptions and patterns are derived and to include demonstrations on both the Nottingham and ECL datasets.
2. A comparison between FSTLLM and the reference paper mentioned in ‘Essential References Not Discussed’ would strengthen the discussion.
3. The rationale for fine-tuning an LLM remains unclear. Given that all domain knowledge is embedded in the prompt, why not leverage powerful LLMs such as GPT-4o or DeepSeek R1 directly for inference-based forecasting enhancement?

**Questions For Authors:**

Please address the three weaknesses listed in the ‘Other Strengths And Weaknesses’ section and all typos listed in ‘Other Comments Or Suggestions’.

**Relation To Broader Scientific Literature:**

This work builds upon recent advancements in using LLMs for forecasting tasks, such as GPT4TS and Time-LLM. While these studies demonstrate the potential of LLMs in processing time-series data, they primarily focus on fine-tuning LLMs with numerical input. FSTLLM advances this line of research by incorporating domain knowledge through structured prompt engineering combined with an STGNN backbone, making LLMs more context-aware for time-series forecasting.

**Theoretical Claims:**

I have checked equations 1 – 8, and they are correct and align with the submitted code.

---

> ### Author Rebuttal · Authors · 2025-03-30
>
> Response to Reviewer
>
> We thank the reviewer for the thoughtful comments. We address each point below and will incorporate the corresponding revisions into the manuscript.
>
> **Q1: Node Description, Pattern Analysis, and Case Study**
>
> We used ChatGPT-4o to generate both node descriptions and pattern analyses. To ensure reproducibility, we will include the prompt templates in the Appendix. Specifically:
>
> * Node Description Prompt:
>
> >“You are given a feature description from [carpark description link] and user reviews for this carpark at [Google review link]. Write an inductive summary about the carpark based on contents from both sources.”
>
> * Node Pattern Analysis Prompt:
>
> > “I will provide one week of parking lot records for a carpark in England. This record is collected every 15 minutes starting from 12:01 PM on 2016-10-26. Please describe the natural pattern of parking availability, including diurnal and weekly variations, and highlight peak and dip periods observed in the data. The records are: [extracted training data].”
>
> Additionally, we have conducted a full fine-tuning and inference case study on the ECL dataset. Due to space limitations in this response, the full results and analysis will be included in the Appendix of the revised manuscript.
>
> **Q2: Discussion of "News2Forecast" (Wang et al., 2024, NeurIPS)**
>
> We will include a discussion of this work in Section 2.3 Related Work, and revise the manuscript as follows:
>
> >“News2Forecast (Wang et al., 2024) enhances time series forecasting by integrating social events through LLM-based agents using reflection and reasoning. It fine-tunes a pre-trained LLM to align textual and numerical data, thereby improving forecasting accuracy.”
>
> While this work focuses on aligning news events with time series data, it does not account for spatial dependencies across different time series. Furthermore, it may suffer from reduced robustness when training samples are limited or when relevant external textual content from news is unavailable. In contrast, FSTLLM explicitly models spatial-temporal dependencies and is designed to generalize under limited supervision.
>
> **Q3: Rationale for Fine-Tuning vs. Direct Prompting with Powerful LLMs**
>
> We appreciate this important question. The motivation for fine-tuning instead of relying solely on inference from powerful LLMs is twofold:
>
> * **Modeling Temporal Dynamics:** Off-the-shelf LLMs (e.g., GPT-4o, DeepSeek R1), when used via prompting, do not effectively model the complex temporal dependencies inherent in time series data. Their predictions tend to be shallow, often yielding weighted approximations based on two provided historical input and numerical prediction, rather than learning temporal patterns explicitly.
>
> * **Learning Temporal Representations via Weight Updates:** Fine-tuning enables the model to update its internal weights, allowing it to capture nuanced dynamics and structure within the time series. This results in significantly improved forecasting performance compared to inference-only approaches.
>
> As evidenced in prior work, fine-tuned models such as GPT4TS and Time-LLM consistently outperform inference-based models like PromptCAST, further justifying our design choice.
>
> Please let us know if further clarification is needed. We are grateful for the reviewer’s constructive feedback.

---

> > ### Comment · Reviewer_KEzm · 2025-04-02
> >
> > I appreciate the author's patient response, which largely addressed my doubts. I will slightly increase my score.

---

> > > ### Author Response · Authors · 2025-04-03
> > >
> > > Dear Reviewer,
> > >
> > > Thank you for taking the time to revisit your review. we truly appreciate your thoughtful reconsideration and the updated score.

---

### Official Review · Reviewer_WS2i · 2025-02-26

**Overall Recommendation:** 4

**Summary:**

This work introduces a framework called FSTLLM. This framework provides enhanced few-shot time series forecasting performance by integrating LLMs with the STGNN backbone. Specifically, it leverages LLMs for spatial correlation modeling, an STGNN network for spatio-temporal pattern modeling, and a domain knowledge injection module for improved predictions. FSTLLM outperforms state-of-the-art baselines, highlighting superior accuracy and robustness in real-world datasets.

**Claims And Evidence:**

The claims in the paper are generally supported by experimental results and case studies.

**Essential References Not Discussed:**

Two important work are missing.

[1] Time-MoE: Billion-Scale Time Series Foundation Models with Mixture of Experts, ICLR 2025.

[2] DUET: Dual Clustering Enhanced Multivariate Time Series Forecasting, KDD 2025.

**Experimental Designs Or Analyses:**

The experimental design and analyses in FSTLLM are methodologically sound, with well-defined datasets, 12 baselines, and standard evaluation metrics (MAE, RMSE, MAPE). The few-shot setting is realistic, using two real-world datasets (Nottingham Parking & ECL) to test generalizability. The analysis demonstrates consistent performance improvements, supported by ablation studies (assessing model components) and a few-shot integration study, showing FSTLLM’s ability to enhance other forecasting models like GPT4TS and iTransformer.

**Methods And Evaluation Criteria:**

Yes. The proposed methods and evaluation criteria are well-suited for few-shot time series forecasting. FSTLLM integrates LLMs for spatial correlation modeling, an STGNN for spatio-temporal pattern modeling, and a domain knowledge injection module for improved predictions. The model is evaluated on real-world datasets (Nottingham Parking and ECL) using standard metrics (MAE, RMSE, MAPE), with a few-shot setup to test its adaptability.

**Other Comments Or Suggestions:**

Typo in the Introduction section, line 36: Missing comma before "offer mechanisms to jointly."

Typo in the Methods section, line 163: "The numerical prediction tokens is still suboptimal" should be "The numerical prediction tokens are still suboptimal."

**Other Strengths And Weaknesses:**

S1. The problem studied in this work (few-shot time series forecasting) is an important field given real-world applications, where data collection is often limited.

S2. The paper is well-written and easy to follow. The experimental analysis illustrates that the proposed FSTLLM is superior to the existing methods.

W1. Terminology Clarity: The term "candidate node embedding" in Section 3.1 (lines 205-206) is unclear. It would help to specify whether it refers to another node in the graph or an alternative embedding technique.

W2. Few-Shot Integration Details: Section 4.4 claims that "we replace the numerical prediction tokens generated by the STGNN backbone with those produced by alternative transformer-based methods." However, implementation details are not clearly specified—particularly whether the LLM-Enhanced Graph Construction Module was considered or omitted, given that GPT4TS does not model multivariate correlations while other transformer-based methods do.

W3. Data Efficiency Evidence: The introduction claims that "these models typically require large volumes of training data...collecting such data is time-consuming and resource-intensive." However, the experimental results primarily focus on forecasting accuracy rather than explicitly demonstrating FSTLLM’s data efficiency. Additional experiments comparing performance with varying data availability could reinforce this claim.

**Questions For Authors:**

I will consider changing my evaluation of the paper based on the author’s response to the following concerns.
1. Discussing Time-MoE in this paper on a few-shot time series forecasting task.
2. Clearly explain what "candidate node embedding" refers to, as listed in W1.
3. Explain the implementation in detail in section 4.4, the Few-Shot Learning Integration Study listed in W2.
4. Discuss and provide experimental results to demonstrate FSTLLM’s data-efficient advantage as listed in W3.

**Relation To Broader Scientific Literature:**

FSTLLM advances spatio-temporal forecasting by leveraging LLMs to enhance spatial correlation modeling, surpassing traditional STGNNs. Unlike existing LLM-based models such as GPT4TS (NeurIPS 2023) and TimeLLM (ICLR 2024), it integrates node-specific domain knowledge for more context-aware predictions, leading to improved forecasting performance. By utilizing LLMs’ few-shot capabilities, FSTLLM enhances forecasting in data-limited settings while also providing reasoning ability, distinguishing it from classical STGNNs like GTS (ICLR 2021) and STEP (KDD 2022). This work introduces a novel approach that combines LLMs with an STGNN backbone rather than relying on a single architecture. Researchers working on other time series tasks like imputation and classification can use this method as a foundation for integrating LLMs with other models.

**Theoretical Claims:**

Yes, the mathematical equations and the claims in the paper appear correct. There are no particular theoretical proofs in the paper.

---

> ### Author Rebuttal · Authors · 2025-03-30
>
> Dear  Reviewer,
>
> We sincerely thank the reviewer for the insightful and constructive feedback. Below, we address each point in detail and outline the corresponding changes we will make in the revised manuscript.
>
> **Q1: Missing References — DUET and Time-MoE**
>
> We appreciate the reviewer’s suggestion and will incorporate both DUET and Time-MoE into our related work discussion:
>
> DUET will be added to Section 2.1 (Classical Neural Network-based Methods), lines 90–93:
>
> >“DUET (Qiu et al., 2025) introduces a framework to tackle multivariate time series forecasting with heterogeneous temporal patterns and complex inter-channel dependencies. It features a Temporal Clustering Module (TCM) for handling temporal heterogeneity, and a Channel Clustering Module (CCM) that applies a novel channel-soft-clustering mechanism in the frequency domain to model inter-channel relationships while mitigating noise.”
>
> Time-MoE will be discussed in Section 2.3 (Large Language Models), lines 126–130:
>
> >“Time-MoE (Shi et al., 2025) presents a scalable architecture for time series forecasting using a sparse mixture-of-experts design to enhance efficiency while maintaining high model capacity. Trained on the extensive Time-300B dataset (over 300 billion time points across nine domains), Time-MoE scales up to 2.4 billion parameters and achieves strong forecasting accuracy. However, its expert routing mechanism is less effective in few-shot scenarios, limiting performance under data-constrained settings.”
>
> **Q2: Clarification of “Candidate Node Embedding”**
>
> Thank you for pointing this out. The term refers to the embedding of a specific candidate node in the graph structure. We will revise the corresponding sentence to reader:
>
> >“... the embedding of a candidate node selected from the spatio-temporal graph.”
>
> This revision will improve clarity and remove ambiguity.
>
> **Q3: Integration Experiment Setup**
>
> We agree that further clarification is needed. We will revise Section 4.4 (Few-Shot Learning Integration Study), lines 407–410, as follows:
>
> >“To assess the plug-and-play potential of our LLM-enhanced graph construction, we substitute the numerical prediction tokens generated by the STGNN backbone with those from various transformer-based forecasting models. Specifically, we remove both the LLM-enhanced graph construction module and the STGNN backbone, and replace them with external transformer-based methods without modifying or retraining those models. This allows us to evaluate the generality of our integration strategy.”
>
> This ensures the experiment strictly evaluates modular compatibility in a plug-and-play fashion.
>
> **Q4: Evidence of Data Efficiency**
>
> Thank you for highlighting this important point. To evaluate data efficiency, we compare FSTLLM trained on just 3 days of data with baselines trained on 30 days of data, using the Nottingham dataset. The results are summarized below:
>
> | Method            | MAE   | RMSE   | MAPE (%) |
> |-------------------|-------|--------|-----------|
> | FSTLLM (3 days)   | 22.84 | 83.68  | 22.33     |
> | GTS (30 days)     | 23.54 | 90.88  | 23.02     |
> | GPT4TS (30 days)  | 27.50 | 86.88  | 25.70     |
> | PatchTST (30 days)| 33.07 | 95.20  | 30.13     |
> | DLinear (30 days) | 32.82 | 92.64  | 26.63     |
>
> These results show that FSTLLM with only 3 days of training data outperforms all baselines trained with 10× more data, highlighting the strong data efficiency of our approach. This table will be added to the Appendix in the revised version.
>
> We appreciate the reviewer’s detailed feedback and believe that the above revisions will substantially improve the clarity and completeness of our submission.

---

### Official Review · Reviewer_vnyj · 2025-02-28

**Overall Recommendation:** 4

**Summary:**

Considering the heavy time cost to collect data for training deep learning time series forecasting models, this work focuses on enhancing forecasting performance with limited training data. This paper propose a graph construction module to ensure stable graph construction used in Spatio-Temporal Graph Neural Networks (STGNNs). Furthermore, it proposed an LLM fine-tuning methodology to enhance forecasting performance with LLM’s embedded knowledge and common-sense reasoning ability. Finally, experiments on two real-world datasets demonstrate this method’s solid and robust performance.

**Claims And Evidence:**

This paper is application-driven, and claims are supported by experimental results.
- Claim 1: The Proposed FSTLLM achieves enhanced few-shot time series forecasting performance. Evidence 1: Supported by experimental results in Table 1 and Table 2.
- Claim 2: The LLM Enhanced Graph Construction module is able to enhance graph construction by embedding contextual information of spatial nodes. Evidence 2: Supported by experimental results in Table 3 from ablation study.
- Claim 3: The Domain Knowledge Injection module enables humanlike consideration and few-shot forecasting performance enhancement. Evidence 3: Supported by experimental results in Table 3 from ablation study and case study visualization in reasoning subsection, section 4.2 Performance of LLM.
- Claim 4: FSTLLM can augment various time series forecasting models, enhancing their performance in few-shot set- tings without updating their parameters. Evidence 4: Supported by experimental results in Table 4 from section 4.4 Few-Shot Learning Integration Study.

**Essential References Not Discussed:**

As far as I know, this work includes the majority of research works related to few-shot time series forecasting from STGNNs, Transformer-based methods, and LLM-based methods. One paper omitted in this work could be ‘AutoTimes: Autoregressive Time Series Forecasters via Large Language Models’ published in NIPPS 2024, which adapts LLM for time series forecasting tasks.

**Experimental Designs Or Analyses:**

The design of the experiments is clear and comprehensive and support FSTLLM’s claims. Details as below. One experimental setting can be discussed to enhance the quality of this work: though using 7 days of data to simulate limited data situations is commonly used, some works also include 3 days of data for comparison as well, such as used in TransGTR (KDD 2023) and CrossTReS (KDD 2022).

**Methods And Evaluation Criteria:**

There are detailed running examples and complexity comparison covering 12 baselines. 2 datasets. 3 metrics. The selection of baselines aligns with domain of time series forecasting. The evaluation criteria, MAE, RMSE, and MAPE are standard evaluation criteria for time series forecasting tasks.

**Other Comments Or Suggestions:**

line 80: "that can commonly used in time series forecasting" → "that are commonly used in time series forecasting"
line 163: "After this, due to the limited training data, the numerical prediction tokens is still suboptimal in capturing temporal dynamics." "is" → "are"

**Other Strengths And Weaknesses:**

Strength.
+ This paper is well written and clear.
+ Extensive experiments, including abundant baselines, real-world datasets, and comprehensive exploration of model performance have been conducted to show the effectiveness of FSTLLM.

Weakness.
- Discussing on experiment setting of 3 days data for comparison as well such as used in TransGTR (KDD 2023) and CrossTReS (KDD 2022).
- Explanation more on integration with existing methods, for instance, which exact components are removed and replaced by the existing methods. Does existing methods need to be re-train in order to suit FSTLLM’s framework.

**Questions For Authors:**

In general, this is an innovative and solid work focus on few-shot time series forecasting task. Serval improvements can be achieved to improve the quality of this paper. These improvements include discussing AutoTimes, discussing experimental settings of using 3 days data to simulate a lack of data, and further explanation on integration experimental details.

**Relation To Broader Scientific Literature:**

The paper contributes to the growing body of research on improving time series forecasting in data-scarce environments compared to methods requiring massive training data such as STGNNs (GraphWaveNet, GTS, STSGCN), Transformer-based methods (PatchTST, iTransformer), and LLM methods (Time-LLM, GPT4TS).

**Theoretical Claims:**

The equations are correct and align with the code submitted. Proposed prompt structure align with the real use case shown in Appendix.

---

> ### Author Rebuttal · Authors · 2025-03-30
>
> Dear Reviewer
>
> We sincerely thank the reviewer for the constructive feedback. We address each of the comments below and will update the manuscript accordingly.
>
> **Q1: Discussion of AutoTimes**
>
> Thank you for the suggestion. We will add the following discussion to Section 2.3 (Related Work):
>
> >"AutoTimes (Liu et al., 2024) repurposes decoder-only large language models (LLMs) for autoregressive time series forecasting by mapping time series inputs into the embedding space of language tokens. This method leverages the sequential modeling strength of LLMs to generate variable-length future predictions without updating the LLM weights."
>
> We acknowledge its relevance and will highlight the distinction between our fine-tuning-based approach and AutoTimes’ frozen-weight generation scheme.
>
> **Q2: 3-Day Few-Shot Forecasting Experiment on Nottingham Dataset**
>
> Due to time constraints, we performed a 3-day training experiment using the Nottingham dataset and a representative subset of baselines. The results are presented below:
>
> | Method     | MAE   | RMSE  | MAPE (%) |
> |------------|-------|--------|-----------|
> | FSTLLM     | 22.84 | 83.68  | 22.33     |
> | GTS        | 29.57 | 87.44  | 26.04     |
> | GPT4TS     | 33.24 | 93.92  | 26.95     |
> | PatchTST   | 34.65 | 97.45  | 32.38     |
> | DLinear    | 37.52 | 95.87  | 30.62     |
>
> These results demonstrate that FSTLLM consistently outperforms strong baselines under limited data scenarios. This further validates the robustness and adaptability of our framework in few-shot settings.
>
> **Q3: Clarification on Integration Experiment Setup**
>
> Thank you for pointing this out. We will revise lines 407–410 in Section 4.4 (Few-Shot Learning Integration Study) to provide a clearer explanation:
>
> >“To assess the plug-and-play potential of our LLM-enhanced graph construction, we substitute the numerical prediction tokens generated by the STGNN backbone with those from various transformer-based forecasting models. Specifically, we remove both the LLM-enhanced graph construction module and the STGNN backbone, and replace them with external transformer-based methods without modifying or retraining those models. This allows us to evaluate the generality of our integration strategy.”
>
> Please let us know if any further clarification is needed. We appreciate your detailed review and helpful suggestions.

---

> > ### Comment · Reviewer_vnyj · 2025-04-03
> >
> > I appreciate the authors' efforts and detailed responses, which have clearly addressed my earlier concerns. I do have one additional query regarding the domain knowledge injection block in FSTLLM. This component seems effective in enhancing forecasting performance through structured fine-tuning. Given this promising result, I'm curious whether integrating domain knowledge from external datasets [1] or from cities with richer data availability [2] could potentially further boost FSTLLM's forecasting accuracy. Have the authors considered exploring this direction? If so, what are their thoughts on its potential benefits?
> >
> > [1] Transferable Graph Structure Learning for Graph-based Traffic Forecasting Across Cities
> >
> > [2] Selective Cross-City Transfer Learning for Traffic Prediction via Source City Region Re-Weighting

---

> > > ### Author Response · Authors · 2025-04-03
> > >
> > > Dear Reviewer,
> > >
> > > Thank you for your insightful question. We agree that incorporating additional data from closely related domains (as described in [1]) could indeed further improve forecasting performance through our domain knowledge injection block. However, our primary objective has been few-shot time series forecasting, and we have thus far limited our scope to in-domain data.
> > >
> > > With respect to works such as  [2] and [3], which transfers knowledge from different time series domains (subway, bike sharing, and ride hailing), we believe their temporal dynamics differ from each other. In our fine-tuning experience, even when input features appear similar, underlying seasonality and demand patterns may vary substantially.  Different to dedicated transfer learning framework that usually design multiple encoder block to process each domain separately, mixing data from such divergent domains in the LLMs fine-tuning stage can introduce domain shift and inconsistencies, thereby risking performance degradation rather than improvement.
> > >
> > > Thank you again for your thorough review and valuable feedback.
> > >
> > > [1] Transferable Graph Structure Learning for Graph-based Traffic Forecasting Across Cities
> > >
> > > [2] Selective Cross-City Transfer Learning for Traffic Prediction via Source City Region Re-Weighting
> > >
> > > [3] Cross-Mode Knowledge Adaptation for Bike Sharing Demand Prediction using Domain-Adversarial Graph Neural Networks

---

### Official Review · Reviewer_jV37 · 2025-03-12

**Overall Recommendation:** 4

**Summary:**

This paper proposes a time-series prediction framework that leverages the prior knowledge of LLMs. Based on Author’s introduction about the framework, the framework can be flexibly applied to any advanced time-series prediction model (such as the STGNNs mentioned in the related works). The experiments were conducted on two datasets—one public and the other self-collected (and will be made public later). The results show that the framework is effective in boosting the state-of-the-art. Overall, this is a novel and effective framework. Although it comes with a certain level of computational cost, it is tolerable, as it only LORA fine-tuning a LLM on a single A6000 GPU. The weaknesses are discussed in the following sections.

**Claims And Evidence:**

The main claim of the paper is that LLMs contain reasoning ability and prior knowledge about locations, which can help with time-series modelling. This claim is based on the widely accepted understanding that LLMs aim to model the world and encode a vast number of everyday concepts and relationships in their hidden states. The authors extract features based on this knowledge. In the experiments, they report the effectiveness of the overall framework, along with ablation studies for each module. The evidence is convincing.

**Essential References Not Discussed:**

The paper lacks sufficient citations related to LLM in-context learning.
In-context learning in LLMs is likely beneficial to the model, as seen from their instruction prompt design. The method requires some preliminary predictions incorporated into the prompts, allowing the LLM to refine and improve them further.

I suggest they cite the following paper to help readers better understand and utilize in-context learning:

Min, S., Lyu, X., Holtzman, A., Artetxe, M., Lewis, M., Hajishirzi, H., & Zettlemoyer, L. (2022). Rethinking the role of demonstrations: What makes in-context learning work?. arXiv preprint arXiv:2202.12837.

**Experimental Designs Or Analyses:**

The authors use one public dataset and one self-collected dataset, with the latter also submitted for review (Nottingham.h5).

There are no obvious issues; the settings (such as evaluate a 15/45/60-mins window) are commonly used in the literature.

**Methods And Evaluation Criteria:**

My simplified understanding of the method is: LLM knowledge is introduced before the input of STGNN, and in-context learning is applied on the output of STGNN, and STGNN serves as the backbone of the framework to achieve robust forecasting performance and reasoning to end users.

The evaluation is primarily based on time-series prediction accuracy compared to previous representative methods. The use of MAE, RMSE, and MAPE are standard, and there is nothing additional to comment on.

**Other Comments Or Suggestions:**

Typo corrections:

In the abstract: “methodologies is” -> “methodologies are”

The authors should proofread the entire paper again.

**Other Strengths And Weaknesses:**

Strengths: Already discussed in the previous sections. The notation is consistent, the experimental results are useful, and the proposed method is sufficiently novel.

Weaknesses:
- The description of Alpha-max is unclear. A value of 1 corresponds to standard SoftMax, while 2 represents a more concentrated distribution.  Is the chosen value <1, 1- 2 or >2?
- The authors need to explicitly describe the role and numerical value of this hyperparameter, as it appears to be important.
- The ways to prepare the node description and node pattern analysis are unclear. The authors need to explain more on this preparation, whether they are directly collected from website or specific data engineering operations needed.

**Questions For Authors:**

Please clarify the issue regarding the Alpha-max parameter and node description preparation mentioned above.

**Relation To Broader Scientific Literature:**

The paper primarily cites relevant works in time-series forecasting and large language models, such as TimeLLM and GPT4TS.

The baselines are up-to-date.

**Theoretical Claims:**

This paper does not focus on theoretical proofs.

---

> ### Author Rebuttal · Authors · 2025-03-30
>
> Dear Reviewer,
>
> We sincerely thank you for your valuable feedback. We address your comments in detail below and will revise the manuscript accordingly:
>
> **Q1. Citation of In-Context Learning**
>
> We appreciate your suggestion regarding the citation of in-context learning. We will revise the manuscript by incorporating appropriate references in lines 57–60, as follows:
>
> > “In contrast, Large Language Models (LLMs) demonstrate strong capabilities in common sense reasoning (Zhao et al., 2023), making them particularly effective for integrating domain-specific and contextual knowledge via in-context learning (Min et al., 2022), in addition to fine-tuning. Furthermore, LLMs exhibit robust performance in few-shot and zero-shot learning scenarios, which are highly relevant for data-scarce forecasting tasks.”
>
> **Q2. Clarification of Alpha-Entmax Value**
>
> Thank you for pointing out the need for clarity regarding the α value in the alpha-entmax transformation. As noted in **Appendix D (Implementation Details)**, we employ an α value of 2.0 in our design. To enhance visibility, we will explicitly state this choice again in **Appendix F (The Alpha-Entmax Function)**, along with a brief justification for its selection.
>
> **Q3. Node Description and Pattern Analysis Methodology**
>
> We appreciate your interest in the generation of node descriptions and pattern analysis. We utilized ChatGPT-4o to assist in these tasks, and we will provide the prompt templates used in the Appendix to ensure reproducibility. Specifically:
>
> * Node Description Prompt:
>
> >“You are given a feature description from [carpark description link] and user reviews from [Google review link]. Please synthesize an inductive description of the carpark using content from both sources.”
>
> * Node Pattern Analysis Prompt:
>
> >“I will provide one week of parking lot records for a carpark in England, recorded every 15 minutes starting from 12:01 PM on 2016-10-26. Please describe the temporal usage patterns of the carpark, including variations throughout the day and across weekdays versus weekends. Indicate observed peak and low-demand periods, along with any consistent behavioral trends in parking availability. The records are: [extracted training data].”
>
> We will incorporate both prompts into the appendix to ensure methodological transparency.
>
> Please let us know if further clarification is needed. Thank you once again for your thoughtful and constructive comments.

---

### Decision · Program_Chairs · 2025-05-01

**Decision:**

Accept (poster)

**Comment:**

The reviewers all consider that the problem studied in this paper is useful, and the proposed method is also novel. The paper also provides convicing experimental results, so I personally recommend and acceptance.